# Non-apoptotic TRAIL function modulates NK cell activity during viral infection

Ludmila Cardoso Alves[1,2], Michael D Berger[1], Thodoris Koutsandreas[3,4], Nick Kirschke[1], Christoph Lauer[1], Roman Spörri[5], Aristotelis Chatziioannou[3,4] (iD), Nadia Corazza[1] & Philippe Krebs[1,*] (iD)

## Abstract

The role of death receptor signaling for pathogen control and infection-associated pathogenesis is multifaceted and controversial. Here, we show that during viral infection, tumor necrosis factor-related apoptosis-inducing ligand (TRAIL) modulates NK cell activity independently of its pro-apoptotic function. In mice infected with lymphocytic choriomeningitis virus (LCMV), *Trail* deficiency led to improved specific CD8[+] T-cell responses, resulting in faster pathogen clearance and reduced liver pathology. Depletion experiments indicated that this effect was mediated by NK cells. Mechanistically, TRAIL expressed by immune cells positively and dose-dependently modulates IL-15 signaling-induced granzyme B production in NK cells, leading to enhanced NK cell-mediated T cell killing. TRAIL also regulates the signaling downstream of IL-15 receptor in human NK cells. In addition, TRAIL restricts NK1.1-triggered IFNγ production by NK cells. Our study reveals a hitherto unappreciated immunoregulatory role of TRAIL signaling on NK cells for the granzyme B-dependent elimination of antiviral T cells.

**Keywords** CD8 T cells; IL-15 signaling; lymphocytic choriomeningitis virus; NK cells; TNF-related apoptosis-inducing ligand
**Subject Categories** Immunology; Microbiology, Virology & Host Pathogen Interaction; Signal Transduction

## Introduction

Natural killer (NK) cells and CD8[+] cytotoxic T cells (CTLs) represent a central line of defense against viral infections through their ability to induce apoptotic death in infected cells. Target cell apoptosis involves exocytosis of cytolytic granules containing perforin and granzymes by cytotoxic lymphocytes [1]. In addition, NK cells and CTLs may trigger apoptosis through the engagement of death receptor ligands such as Fas ligand (FasL), tumor necrosis factor (TNF), or TNF-related apoptosis-inducing ligand (TRAIL) on cognate receptors expressed on target cells [2–5]. Death receptor-mediated apoptosis exerts distinct effects on viral replication and infection-associated pathogenesis [5–7]. In particular, the role of TRAIL during infection is controversial. TRAIL is selectively upregulated on influenza-specific CD8[+] T cells to eliminate virus-infected alveolar epithelial cells [2]. TRAIL expression on NK cells also appears to limit the *in vivo* replication of encephalomyocarditis virus [8]. However, TRAIL leads to severe inflammation and tissue damage in *Listeria*-infected wild-type (WT) mice, which is associated with impaired pathogen clearance and reduced survival compared with *Tnfsf10/Trail⁻/⁻* counterparts [9]. Moreover, NK cells induce TRAIL signaling to eliminate hepatitis B virus (HBV)-specific CD8[+] T cells [10] or activated CD4[+] T cells during chronic murine cytomegalovirus (MCMV) infection [11], thereby negatively regulating antiviral immunity. A comparable immunomodulatory effect of TRAIL has also been reported in rheumatoid arthritis, where TRAIL signaling limits pathology and inflammation, possibly independently of its pro-apoptotic function [12,13]. Indeed, such non-apoptotic (non-canonical) TRAIL activity has been mainly reported in cancer, with TRAIL signaling exerting a pro-tumorigenic effect in resistant tumor cells [14]. Non-canonical TRAIL signaling in cancer cells elicits receptor-induced kinase activation triggering survival, proliferation, migration, and metastasis [15,16]. Although the ability of TRAIL to induce such non-canonical signaling in malignant cells is well established, it is currently unclear whether TRAIL may exert a non-apoptotic function in non-transformed cells, in particular in the context of viral infection.

Here, we investigated the contribution of TRAIL to the immune response induced by lymphocytic choriomeningitis virus (LCMV). We found an increased specific CD8[+] T-cell response and a faster virus control in infected *Trail⁻/⁻* versus WT mice. This phenotype was ascribed to a reduced ability of NK cells to limit LCMV-specific CD8[+] T cells in *Trail⁻/⁻* mice. Further mechanistic studies revealed that *Trail* blockade mitigated the IL-15 signaling-induced granzyme B production in NK cells in a cell-extrinsic and dose-dependent manner—thereby accounting for the reduced T-cell killing. In addition, TRAIL signaling in NK cells repressed IFNγ production induced

1  Institute of Pathology, University of Bern, Bern, Switzerland
2  Graduate School for Cellular and Biomedical Sciences, University of Bern, Bern, Switzerland
3  Institute of Biology, Medicinal Chemistry & Biotechnology, NHRF, Athens, Greece
4  e-NIOS PC, Kallithea-Athens, Greece
5  Institute of Microbiology, ETH Zurich, Zurich, Switzerland
*Corresponding author. Tel: +41 31 632 4971; Fax: +41 31 381 8764; E-mail: philippe.krebs@pathology.unibe.ch

upon NK1.1 receptor activation. Taken together, these results unveil a previously unappreciated regulatory role of TRAIL for NK cell function during infection, which is independent of TRAIL pro-apoptotic activity.

# Results

## LCMV-infected $Trail^{-/-}$ mice show increased CD8$^+$ T-cell response and improved virus clearance

To study the impact of TRAIL on virus immunity, we analyzed virus-specific CD8$^+$ T-cell responses in WT and $Trail^{-/-}$ mice after infection with LCMV strain WE. The frequencies (Fig 1A) and total numbers (Fig EV1A) of IFN$\gamma$-positive and IFN$\gamma$/TNF double-positive CD8$^+$ T cells specific for the LCMV glycoprotein (GP) epitope GP$_{33-41}$ were increased in spleens and livers of $Trail^{-/-}$ mice compared with control animals. We also observed higher frequencies of cytokine-producing CD8$^+$ T cells specific for the H-2D$^b$-restricted nucleoprotein (NP) epitope NP$_{396-404}$ in $Trail^{-/-}$ mice (Fig EV1B). No differences in CD4$^+$ T cells specific for the GP epitope GP$_{61-80}$ were detected (Fig EV1C), indicating that TRAIL exerts a CD8$^+$ T cell-restricted effect on the LCMV-specific T-cell response. $Trail^{-/-}$ mice also showed reduced CD8$^+$ T cell-mediated liver immunopathology, as measured by lower levels of serum alanine transaminase (ALT) (Fig 1B).

As CD8$^+$ T cells are crucial for LCMV clearance [17], we next analyzed virus titers in spleen and liver at different time points. No differences in virus titer were observed until 8 days of infection, suggesting that TRAIL does not affect early control of LCMV-WE. However, $Trail^{-/-}$ mice showed accelerated virus clearance compared with WT animals, with complete virus elimination in spleen and liver 12 days after LCMV infection (Fig 1C).

In the following, we addressed whether TRAIL expression on CD8$^+$ T cells may directly contribute to the LCMV-specific CD8$^+$ T-cell response. To this end, we transferred congenic $Trail^{+/+}$ T-cell receptor (TCR) transgenic CD8$^+$ T cells specific for the LCMV glycoprotein GP$_{33-41}$ (P14 cells) into WT and $Trail^{-/-}$ mice previously infected with LCMV (Fig 1D). Under these conditions, P14 cells primed in $Trail^{-/-}$ recipients expanded at higher frequencies (Fig 1E) and produced more inflammatory cytokines (Fig 1F).

Taken together, these data reveal that TRAIL limits the expansion of LCMV-specific CD8$^+$ T cells in a cell-extrinsic manner, thereby modulating virus clearance and liver immunopathology.

## The increased specific CD8$^+$ T-cell response in LCMV-infected $Trail^{-/-}$ mice depends on NK cells

To further investigate the role of $Trail$ for the LCMV-specific immune response, we assessed the kinetics of $Trail$ expression in infected mice. There was a substantial increase in $Trail$ transcripts in spleen and liver in the first days of infection, which then progressively declined to naïve levels after 8 days (Fig 2A), thus suggesting a contribution of TRAIL early during LCMV infection. We next measured inflammatory cytokines released systemically to identify immune populations that were possibly altered in recently infected $Trail^{-/-}$ versus WT animals. Among the cytokines analyzed, we found in $Trail^{-/-}$ mice threefold higher serum levels of IFN$\gamma$

(Figs 2B and EV1D), a cytokine that is rapidly secreted by innate lymphocytes, in particular NK cells, following viral infection [18]. Indeed, early LCMV infection triggered an upregulation of TRAIL and TRAIL receptor (TRAIL-R or DR5, which is encoded by $Tnfrsf10b$) on NK cells (Fig 2C), thus indicating a potential effect of TRAIL/TRAIL-R signaling on NK cell activity during LCMV infection. Of note, other immune cell populations including CD3$^+$, CD4$^+$, or CD8$^+$ T cells; dendritic cells (DCs, defined as CD11c$^+$MHCII$^+$Ly6C$^-$Ly6G$^-$, CD11c$^+$MHCII$^+$Ly6C$^-$Ly6G$^-$CD8$^+$, or CD11c$^+$MHCII$^+$Ly6C$^-$Ly6G$^-$CD8$^-$ cells); or neutrophils (defined as CD11b$^+$CD11c$^-$Ly6G$^+$) did not express DR5 24 h after infection, while monocytes (defined as CD11b$^+$CD11c$^-$Ly6C$^+$Ly6G$^-$ cells) upregulated DR5 (Fig EV1E).

Previous reports have shown that besides their well-described antiviral and anti-tumor functions, NK cells can also regulate T-cell responses and thereby influence the outcome of viral infections [11,19–21]. To address a possible role of these cells in our model, we next depleted NK1.1$^+$ cells in WT and $Trail^{-/-}$ mice before infection with LCMV-WE. In line with previous studies [19–21], antibody-mediated NK cell depletion increased the LCMV-specific CD8$^+$ T-cell response in WT mice. However, NK cell depletion also abrogated the limiting effect of $Trail$ on T-cell priming (Fig 2D), and it comparably prevented liver immunopathology in WT and $Trail^{-/-}$ mice (Fig 2E). The virus-specific CD4$^+$ T-cell response was also enhanced in NK cell-depleted animals, yet to the same extent in the two mouse strains (Fig EV1F).

These results indicate that, during LCMV-WE infection, $Trail$ contributes to the NK cell-mediated regulation of the specific CD8$^+$ T-cell response.

## $Trail$ controls cytokine production in NK cells during LCMV-WE infection

We next applied flow cytometry to determine whether NK cells were the source of higher serum IFN$\gamma$ in LCMV-infected $Trail$ mice. The frequencies and numbers of IFN$\gamma$-positive NK cells were increased in the spleens and livers of $Trail^{-/-}$ versus WT mice, most prominently 24 h after infection (Fig 3A and B). In addition, $Trail^{-/-}$ NK cells expressed higher levels of IFN$\gamma$ (Fig 3C). These data on altered IFN$\gamma$ in the NK cells of infected $Trail^{-/-}$ mice were further corroborated by a gene expression analysis, revealing that several pathways related to inflammation and cytokine production or signaling are differently affected in $Trail^{-/-}$ versus WT NK cells upon LCMV infection (Figs EV2A–C and EV3). This was associated with a threefold rise in serum IFN$\gamma$ in infected $Trail^{-/-}$ animals, which could be abrogated by NK cell depletion (Fig 3D). Yet, this increase in serum IFN$\gamma$ was not due to alterations in absolute numbers of NK cells in spleens and livers of $Trail^{-/-}$ mice compared with WT counterparts (Fig EV4A).

NK cell-secreted IFN$\gamma$ induces the maturation of DCs, leading to enhanced CD8$^+$ T-cell priming [22,23]. However, we did not find differences in cell frequencies, total numbers or expression of activation markers for DCs in WT and $Trail^{-/-}$ animals 24 h after infection (Fig EV4B–E). Therefore, we concluded that DCs likely did not account for the altered response to LCMV-WE in $Trail^{-/-}$ mice.

Alternatively, IFN$\gamma$ has also been reported to directly promote T-cell responses [24,25]. Therefore, to investigate whether NK cell-produced IFN$\gamma$ contributed to T-cell activation in our model, we

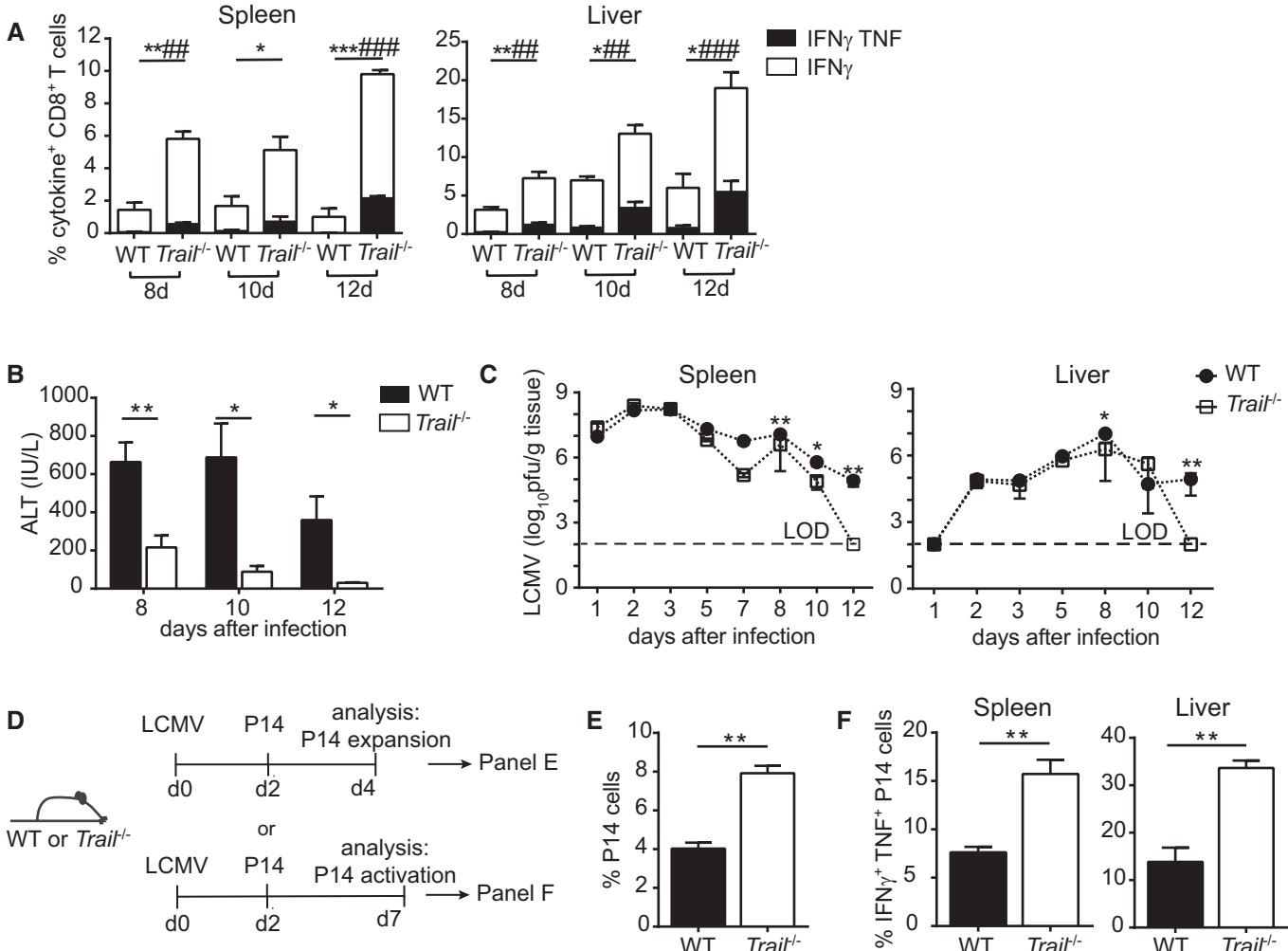

**Figure 1. Altered response to LCMV in *Trail^−/−* mice.**

A Frequencies of cytokine-producing GP$_{33-41}$-specific CD8$^+$ T cells were measured on days 8, 10, and 12 after LCMV infection; *n* = 3 mice per group per day. Data indicate mean ± SEM and show one representative of at least 2–4 experiments. Statistical analyses were performed using unpaired two-tailed *t*-test. *P < 0.05; **P < 0.01; ***P < 0.001 between IFNγ$^+$ cells. ##P < 0.01; ###P < 0.001 between IFNγ$^+$ TNF$^+$ cells.

B Liver immunopathology was assessed by measuring serum ALT at the indicated time points post-LCMV infection. Data shown are mean ± SEM of *n* = 6 mice per group per day, pooled from two independent experiments. Statistical analyses were performed using unpaired two-tailed *t*-test. *P < 0.05; **P < 0.01.

C Virus titers were determined in spleen and liver; *n* = 3–11 mice per group per time point. Dotted horizontal lines indicate the limit of detection (LOD). Data were pooled from 1 to 4 independent experiments. Statistical analyses were performed using Mann–Whitney test. *P < 0.05; **P < 0.01.

D, E Experimental setup of P14 cell transfer experiments following LCMV infection of WT and *Trail^−/−* mice (D). P14 cell expansion was analyzed in spleen 4 days postinfection; *n* = 3 mice per group (E). Data indicate mean ± SEM and show one representative of at least 2–4 experiments. Statistical analyses were performed using two-tailed *t*-test. **P < 0.01.

F Frequencies of IFNγ$^+$ TNF$^+$ P14 cells were measured in the indicated organs 7 days postinfection with LCMV; *n* = 3 mice per group. Data indicate mean ± SEM and show one representative of at least 2–4 experiments. Statistical analyses were performed using unpaired two-tailed *t*-test. **P < 0.01.

depleted NK cells immediately before or early after LCMV infection and analyzed specific CD8$^+$ T cells (Fig 3E). This approach allowed discriminating possible effects of NK cells versus NK cell-derived early IFNγ during T-cell priming, respectively. Whereas both NK cell depletion regimens improved the virus-specific T-cell response 8 days after LCMV infection, they also both abolished the differences between WT and *Trail^−/−* animals (Fig 3F).

Taken together, these data suggest that the augmented CD8$^+$ T-cell priming in *Trail^−/−* mice is independent of their initial elevated level of systemic IFNγ.

### *Trail^−/−* NK cell shows impaired cytotoxicity associated with reduced granzyme B expression

Besides cytokine production early during infection, NK cells may also regulate specific T-cell responses through their cytolytic activity [19–21,26,27]. To assess the cytotoxicity of WT versus *Trail^−/−* NK cells after LCMV infection, we next performed an *in vitro* killing assay using TRAIL-resistant YAC-1 cells [28] as NK cell targets. *Trail^−/−* splenic NK cells isolated from LCMV-infected animals showed reduced cytotoxicity compared with WT NK cells (Fig 4A). Using

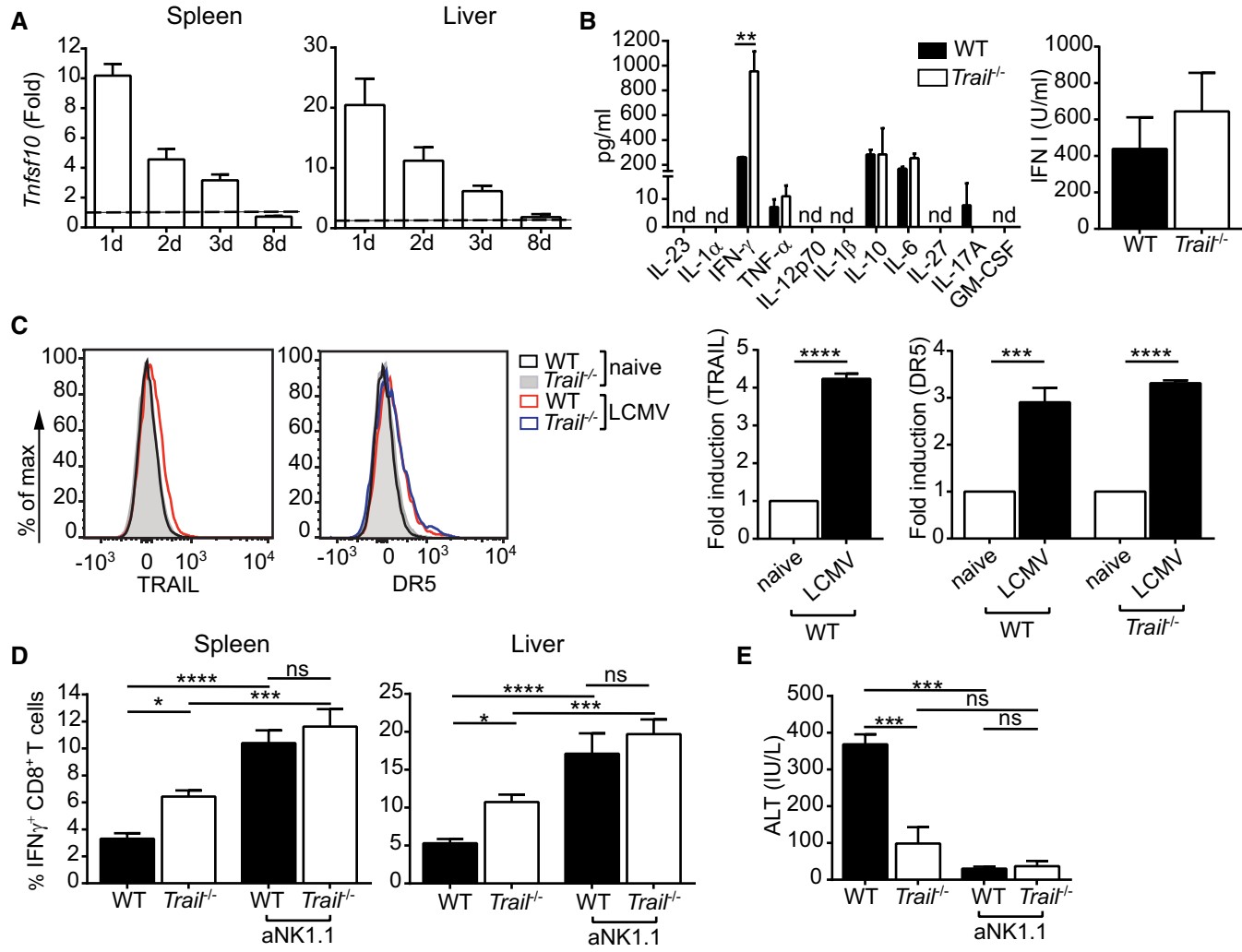

**Figure 2. NK cells contribute to the altered CD8⁺ T-cell response in infected *Trail⁻/⁻* mice.**

A  WT mice were infected with LCMV, and *Tnfsf10/Trail* transcript levels were measured in spleen and liver at the indicated time points. Data are represented as fold induction after normalization to levels in naïve tissue and are mean ± SEM of *n* = 3–9 mice per group per time point, pooled from two independent experiments.

B  Mice were infected with LCMV, and cytokine concentrations (left panel) or type I IFN activity (right panel) was measured in the serum after 24 h using a cytokine multiplex assay or a bioassay, respectively. Data indicate mean ± SEM of *n* = 4 mice per group. nd, non-detectable. One experiment was performed. Statistical analyses were performed using unpaired two-tailed *t*-test. **P < 0.01.

C  TRAIL and TRAIL-R (DR5) surface protein expression were measured by flow cytometry on splenic NK cells. Representative histograms (left-hand side) and corresponding fold increase of mean fluorescence intensity (right-hand side) are depicted. Values shown were normalized to naive controls. Data indicate mean ± SEM of *n* = 4 for groups of infected mice. One experiment was performed. Statistical analyses were performed using one-sample *t*-test. ***P < 0.001; ****P < 0.0001.

D  Frequencies of IFNγ⁺ GP₃₃₋₄₁-specific CD8⁺ T cells were measured in the indicated organs 8 days postinfection. When indicated, NK cells were depleted (aNK1.1). Data indicate mean ± SEM of *n* = 3 for spleen and *n* = 6 for liver. One representative of three independent experiments is shown. Statistical analyses were performed using one-way ANOVA with Tukey post-test. Only the indicated groups were compared for statistical analysis. ns, non-significant; *P < 0.05; ***P < 0.001; ****P < 0.0001.

E  Serum ALT was measured in the indicated groups of mice 8 days postinfection. Data shown are mean ± SEM of *n* = 3 mice per group. One representative of three independent experiments is shown. Statistical analyses were performed using one-way ANOVA with Tukey post-test. Only the indicated groups were compared for statistical analysis. ns, non-significant; ***P < 0.001.

*Ifnar1⁻/⁻* cells as *in vivo* targets, which are particularly susceptible to perforin/granzyme-triggered NK cell-mediated lysis [29,30], we also found that the NK cell-mediated elimination of antigen-specific T cells was reduced in LCMV-infected *Trail⁻/⁻* versus WT mice (Fig 4B and C). The underlying mechanisms appeared to be independent of TRAIL pro-apoptotic function since virus-specific CD8⁺ T cells do not

express DR5 in LCMV-infected mice (Fig 4D). The altered cytotoxicity of *Trail⁻/⁻* NK cells was also not due to impaired degranulation, as assessed by surface CD107a expression (Fig 4E), but rather linked to reduced levels of granzyme B protein (Fig 4F).

Importantly, *Trail⁻/⁻* NK cells in LCMV-infected mice showed WT levels of several activating or inhibiting markers including

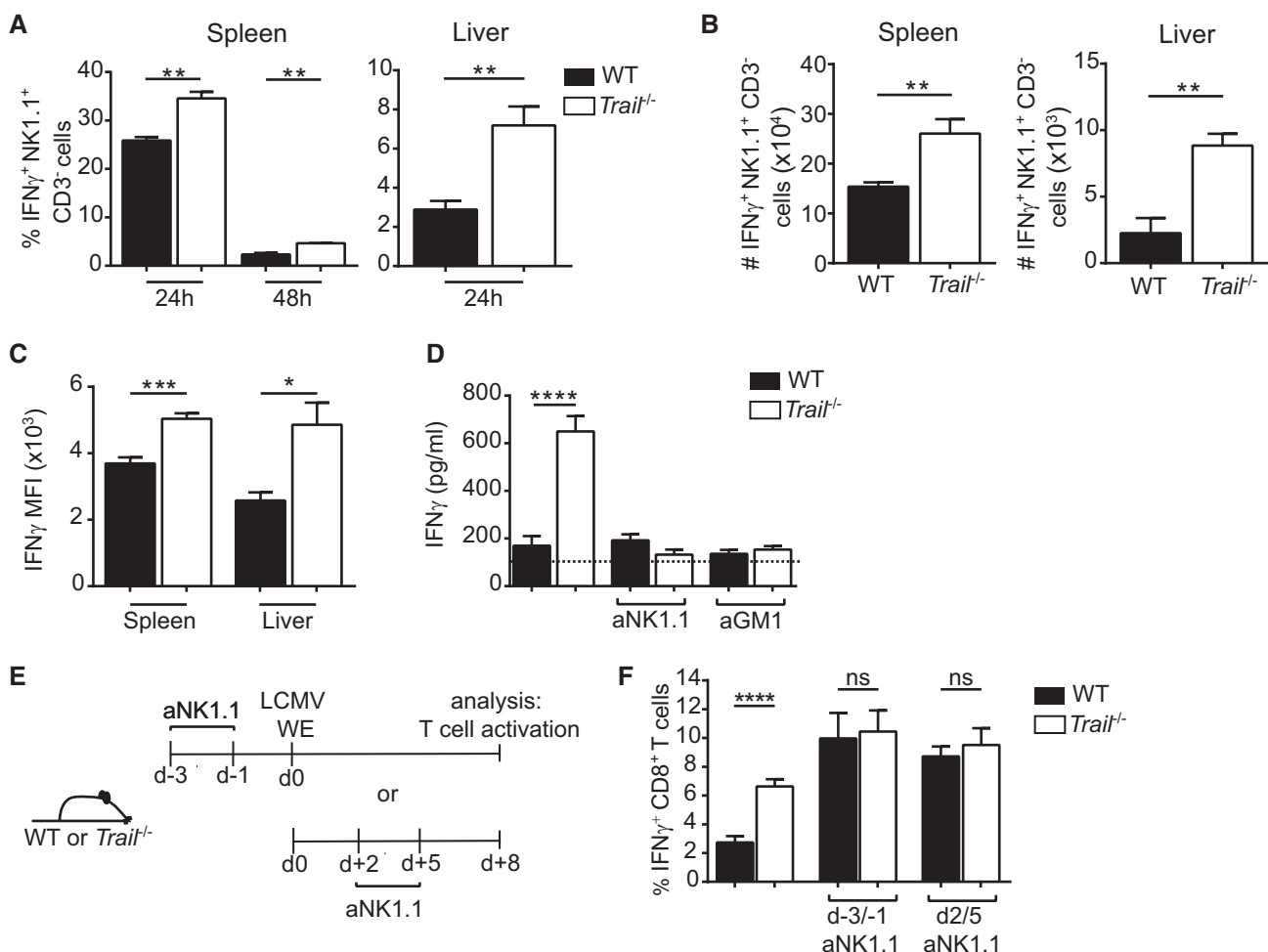

**Figure 3. Enhanced IFNγ production in *Trail*[−/−] NK cells does not directly modulate the specific CD8[+] T-cell response.**

A    Frequencies of IFNγ-positive NK cells were measured 24 and 48 h post-LCMV infection in spleen and 24 h postinfection in liver. Data indicate mean ± SEM of $n = 3$ mice per group and show one representative of at least two independent experiments. Statistical analyses were performed using unpaired two-tailed *t*-test. **$P < 0.01$.

B–D  Total numbers of IFNγ-positive NK cells (B), IFNγ expression levels in NK cells (C) and serum IFNγ levels (D) were measured 24 h postinfection. Where indicated, mice were pretreated 3 and 1 day before infection with an anti-NK1.1 antibody (aNK1.1) or 1 day before infection with an anti-asialo GM1 antibody (aGM1). MFI, mean fluorescence intensity. In (D), baseline IFNγ serum levels in naïve mice ($n = 4$) are indicated by a dotted line. Data indicate mean ± SEM of $n = 3$ (B, C) and $n = 6$ (D) mice per group and show one representative of at least three independent experiments. Statistical analyses were performed using unpaired two-tailed *t*-test. *$P < 0.05$; **$P < 0.01$; ***$P < 0.001$; ****$P < 0.0001$.

E, F  Experimental setup of aNK1.1 treatment before or after LCMV infection (E). Frequencies of cytokine-producing GP$_{33-41}$-specific CD8[+] T cells were measured 8 days postinfection in the indicated groups of control or aNK1.1-treated mice (F). Data indicate mean ± SEM of $n = 6$ mice per group and show one representative of at least two independent experiments. Statistical analyses were performed using unpaired two-tailed *t*-test. Only the indicated groups were compared for statistical analysis. ns, non-significant; ****$P < 0.0001$.

CD69, thus suggesting that these cells had been properly activated. Only the activating receptor Ly49H was reduced by 20 and 31.7% on splenic and hepatic NK cells of infected *Trail*[−/−] mice, respectively (Fig EV4F and G). Yet, Ly49H levels were also diminished on NK cells from naïve *Trail*[−/−] animals (Fig EV4H), implying that LCMV infection did not contribute to this reduced expression.

Furthermore, T-bet (*Tbx21*) and Eomesodermin (*Eomes*), two transcriptional regulators of NK cell development, maturation and function [31], were comparably expressed in NK cells from WT and *Trail*[−/−] infected mice (Fig EV4I and J). Thus, these results suggest that these transcription factors likely do not play a role for the decreased cytolytic activity of *Trail*[−/−] NK cells.

Therefore, our findings indicate that the reduced cytotoxicity of NK cells likely underlies the increased specific CD8[+] T-cell response in LCMV-infected *Trail*[−/−] mice.

## Impaired IL-15 signaling contributes to reduced granzyme B production in *Trail*[−/−] NK cells

Resting murine NK cells express high levels of granzyme B transcripts. Upon activation, granzyme B protein expression is strongly enhanced in NK cells, yet with minimal changes in granzyme B mRNA abundance [32]. To investigate a potential effect of TRAIL on granzyme B transcription, we measured *Gzmb* expression in naïve

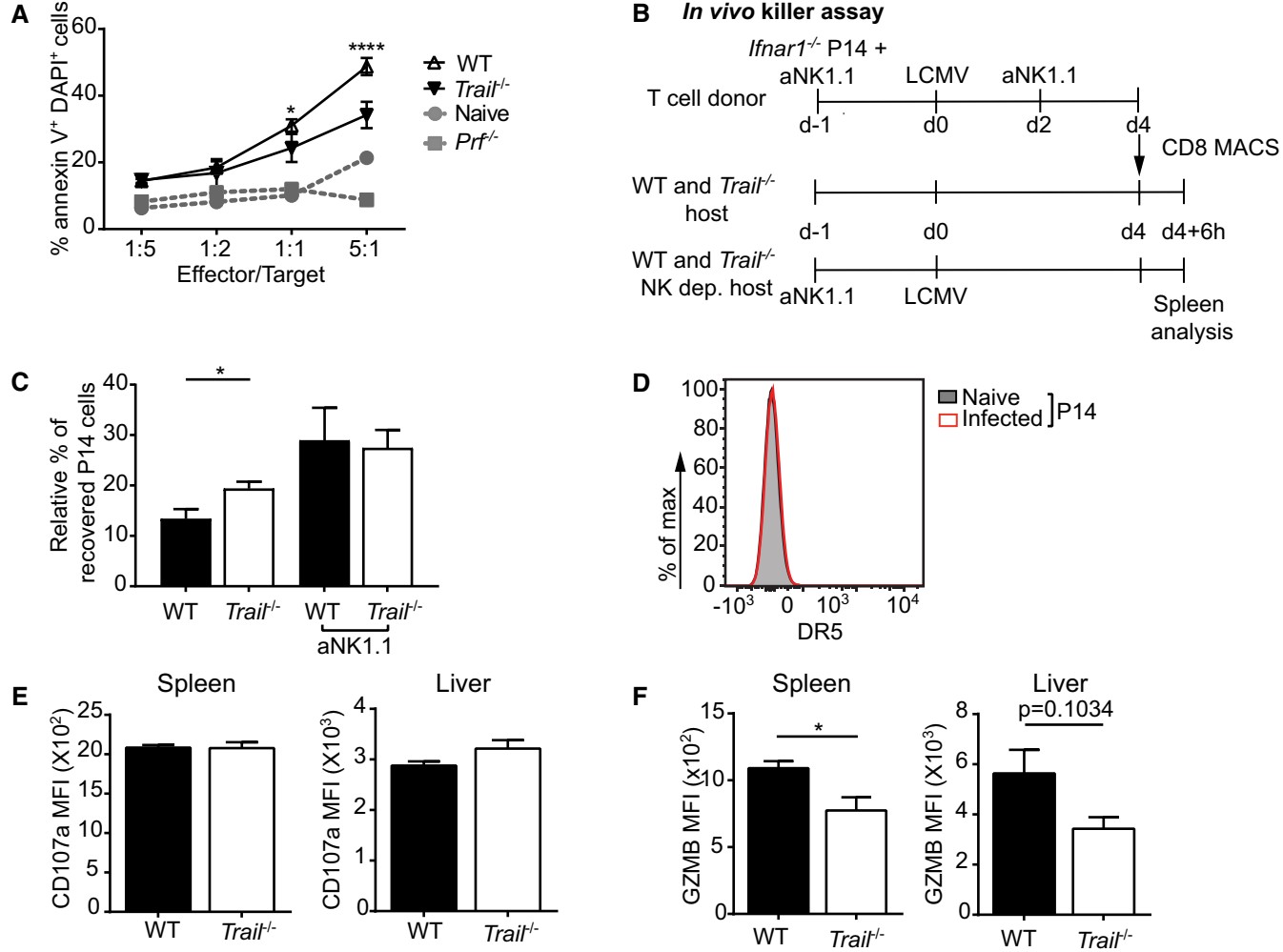

**Figure 4. Impaired cytotoxicity of *Trail*⁻/⁻ NK cells is linked to reduced GZMB production.**

A  WT and *Trail*⁻/⁻ mice were infected with LCMV, and NK cells were analyzed 5 days postinfection. Splenocytes were incubated with TRAIL-resistant YAC-1 cells at the indicated effector/target ratios. Frequencies of annexin V- and DAPI-double-positive cells are indicated. *Prf*⁻/⁻ indicates splenocytes from infected perforin-deficient mice. Data indicate mean ± SEM of *n* = 3 mice per group and show one representative of three independent experiments. Statistical analyses were performed using two-way ANOVA with Tukey post-test. *P < 0.05; ****P < 0.0001.

B–D  Experimental setup of *in vivo* NK cytotoxicity assay using *Ifnar1*⁻/⁻ P14 cells as target cells (B). Frequencies of *Ifnar1*⁻/⁻ P14 cells were measured in the spleen 6 h after adoptive transfer. Data indicate mean ± SEM of *n* = 3 mice per group (C). DR5 expression was assessed on adoptively transfer P14 cells 6 days after LCMV infection (D). Naïve P14 cells were used as a control (*n* = 3 mice). (C, D) One experiment was performed. Statistical analyses were performed using unpaired two-tailed *t*-test. *P < 0.05.

E  WT and *Trail*⁻/⁻ mice were infected with LCMV, and NK cells were analyzed 5 days postinfection. CD107a expression on splenic and hepatic NK cells was measured 5 h after incubation with YAC-1 cells seeded at an effector/target ratio of 1:1. Data indicate mean ± SEM of *n* = 3 mice per group and show one representative of three independent experiments. MFI, mean fluorescence intensity. Statistical analyses were performed using unpaired two-tailed *t*-test.

F  WT and *Trail*⁻/⁻ mice were infected with LCMV, and NK cells were analyzed 5 days postinfection. Splenic and hepatic NK cells were stained for granzyme B (GZMB). Data indicate mean ± SEM of *n* = 3 mice per group and show one representative of at least two independent experiments. MFI, mean fluorescence intensity. Statistical analyses were performed using unpaired two-tailed *t*-test. *P < 0.05.

NK cells from spleen and bone marrow. There were comparable levels of *Gzmb* transcripts in naïve *Trail*⁻/⁻ versus WT NK cells, indicating that *Trail* deficiency does not affect constitutive *Gzmb* expression (Fig EV4K). In agreement with these data, frequencies of CD11b^high^CD27^low^ NK cells, which upregulate cytotoxicity-related transcripts [33], were unchanged in naïve *Trail*⁻/⁻ mice (Fig EV4L).

Granzyme B protein production in NK cells is induced by engagement of IL-15/IL-15 receptor (IL-15R) signaling [32]. To address a possible defect in this signaling pathway, we first measured *Il15* and

IL-15Rβ (CD122) expression during LCMV infection. We found comparable *Il15* transcript levels in spleen and liver tissues of WT and *Trail*⁻/⁻ mice 24 h after LCMV infection (Fig 5A). In addition, there was no difference in surface expression of IL-15β receptor on *Trail*⁻/⁻ versus WT NK cells, implying intact ability for these cells to bind trans-presented IL-15 (Fig 5B and C).

IL-15/IL-15R signaling is conveyed through the PI3K-AKT-mTOR pathway to induce granzyme B expression in NK cells [34]. Flow cytometry analysis of splenocytes isolated 24 h after LCMV infection

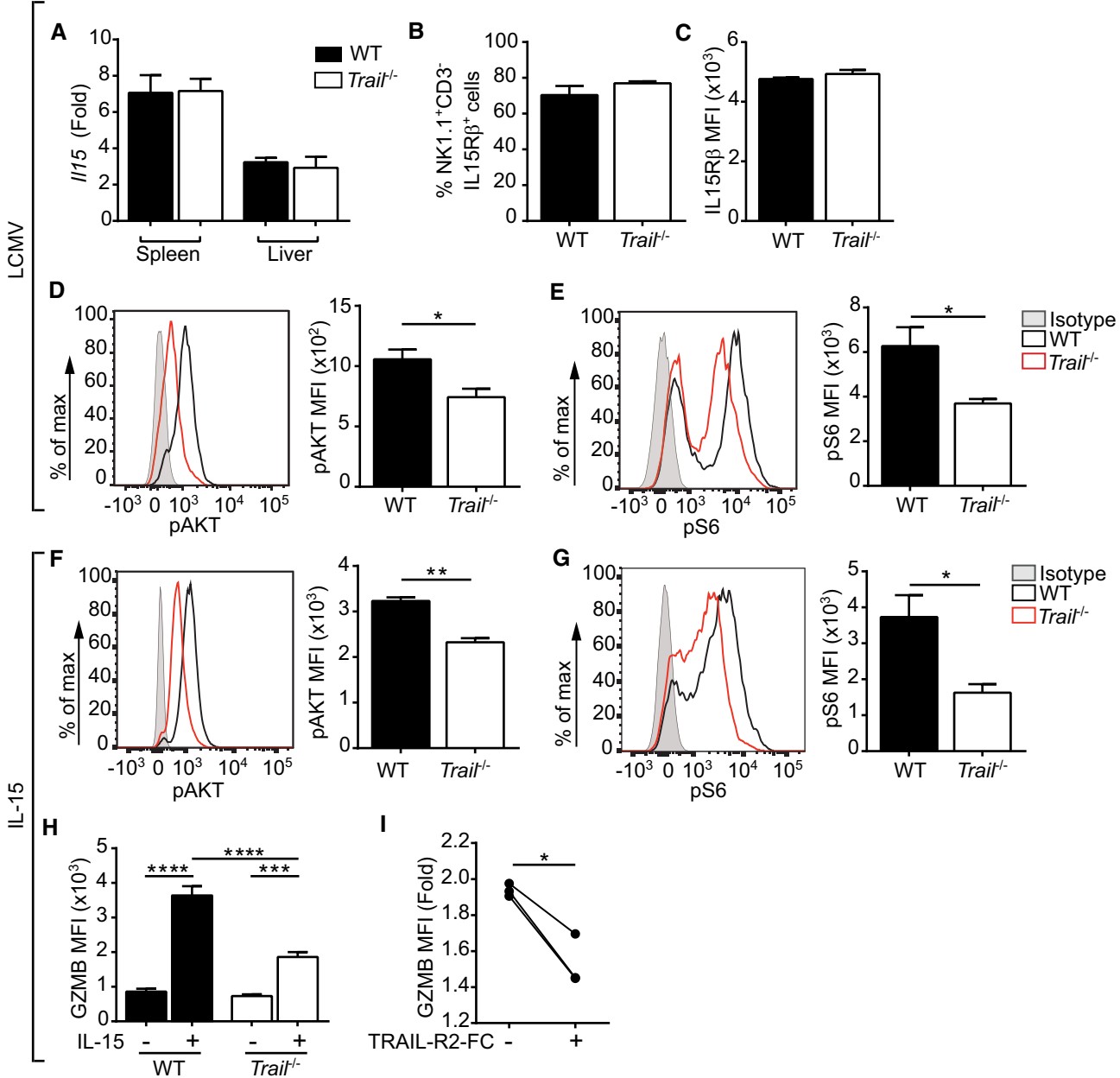

**Figure 5. IL-15 signaling-dependent GZMB pathway is defective in *Trail*<sup>−/−</sup> NK cells.**

A   Mice were infected with LCMV, and *Il15* transcript levels were measured in the indicated organs 24 h postinfection. Data are represented as fold induction after normalization to levels in corresponding naïve tissue. Data indicate mean ± SEM of *n* = 3 mice per group and show one representative of two independent experiments. Statistical analyses were performed using unpaired two-tailed *t*-test.

B–E   Mice were infected with LCMV, and flow cytometry was applied on splenic NK cells from infected animals to assess frequencies of IL-15Rβ-positive cells (B), IL-15Rβ expression levels (C), AKT phosphorylation (D), or S6 phosphorylation (E). For AKT and S6 phosphorylation, representative histograms and cumulative results are depicted. MFI, mean fluorescence intensity; Isotype, isotype-matched control antibody. Data indicate mean ± SEM of *n* = 3 (B–D) or *n* = 4 (E) mice per group and show one representative of at least three independent experiments. Statistical analyses were performed using unpaired two-tailed *t*-test. *P < 0.05.

F, G   Splenic NK cells from naïve donors were stimulated *in vitro* for 1 h with IL-15, and phosphorylation of AKT (F) or S6 (G) was measured. Data indicate mean ± SEM of *n* = 3 (F) or *n* = 4 (G) mice per group and show one representative of at least three independent experiments. Statistical analyses were performed using unpaired two-tailed *t*-test. *P < 0.05; **P < 0.01.

H   GZMB expression was measured in splenic NK cells after stimulation with IL-15 for 20 h. Data indicate mean ± SEM of *n* = 4 mice per group and show one representative of at least three independent experiments. Statistical analyses were performed using one-way ANOVA with Tukey post-test. ***P < 0.001; ****P < 0.0001.

I   WT splenocytes were cultured with IL-15 ± TRAIL-R2-Fc chimeric protein, and GZMB expression was measured in NK cells. Values shown were normalized to unstimulated controls. Data show *n* = 3 mice per group and show one representative of at least three independent experiments. Statistical analyses were performed using paired two-tailed *t*-test. *P < 0.05.

revealed decreased phosphorylation of AKT and S6—two markers of PI3K-AKT-mTOR pathway activation—in NK cells from *Trail*$^{-/-}$ compared with WT mice (Fig 5D and E). These findings were further substantiated by a transcriptomic analysis, indicating that several pathways related to PI3K/AKT signaling and IL-2 family signaling (to which IL-15 belongs) are differently affected in WT versus *Trail*$^{-/-}$ NK cells upon LCMV infection (Tables 1 and 2).

To confirm that these differences were associated with IL-15/IL-15R signaling and not due to confounding factors related to LCMV infection, we next assessed the ability of *Trail*$^{-/-}$ NK cells to transduce IL-15 signals *in vitro*. We indeed found diminished phosphorylation of AKT (Fig 5F) and S6 (Figs 5G and EV5A) in IL-15-stimulated NK cells from *Trail*$^{-/-}$ versus WT mice. This resulted in reduced granzyme B levels in these cells (Fig 5H). Of note, basal levels of phosphorylated AKT and S6 were comparable in naïve *Trail*$^{-/-}$ versus WT NK cells (Fig EV5B). Moreover, pharmacological inhibition of PI3K reduced granzyme B production in IL-15-stimulated WT NK cells to levels observed in *Trail*$^{-/-}$ counterparts, thereby suggesting a link between TRAIL signaling and PI3K activation in NK cells (Fig EV5C).

To establish a direct involvement of TRAIL for downstream IL-15/IL-15R signaling and granzyme B expression, we next addressed the effect of simultaneous TRAIL blockade and IL-15 activation. Such combined treatment led to a reduction in the phosphorylation of AKT (Fig EV5D) and S6 (Fig EV5E), and a consequent diminished granzyme B expression in treated WT NK cells (Fig 5I).

Importantly, TRAIL blockade of activated human NK cells also repressed S6 phosphorylation downstream of IL-15 receptor (Fig 6). Although this effect was less pronounced than in murine NK cells, these results hint at similar TRAIL signaling-dependent regulatory mechanisms in NK cells of both species.

As our transcriptomic analysis had disclosed multiple pathways related to inflammation to be differently affected in *Trail*$^{-/-}$ versus WT NK cells upon LCMV infection, we then also investigated the mechanisms underlying the increased IFNγ levels in NK cells of infected *Trail*$^{-/-}$ mice. IFNγ in NK cells can be induced following stimulation with IL-12 or IL-18, which exhibit synergistic effects [35]. Alternatively, IFNγ may be produced downstream of activating NK receptors [36,37]. Following incubation of WT and *Trail*$^{-/-}$ splenocytes with IL-12 or IL-18, we did not find differences in the frequency of IFNγ-positive NK cells (Fig EV5F) or in the amount of IFNγ produced (Fig EV5G and H). In contrast, splenocyte activation by NK1.1 crosslinking resulted in higher percentages of IFNγ-positive *Trail*$^{-/-}$ than WT NK cells (Fig EV5I), yet comparable granzyme B expression (Fig EV5J). These results indicate that *Trail* deficiency promotes NK1.1 receptor-induced NK cell activation.

Taken together, these findings reveal that TRAIL promotes IL-15 signaling-induced granzyme B production in NK cells. The impaired expression of granzyme B in *Trail*-deficient NK cells is associated with a reduced cytotoxicity, which likely accounts for the improved virus-specific CD8$^+$ T-cell response observed in LCMV-infected *Trail*$^{-/-}$ mice. In addition, TRAIL restricts NK1.1-induced IFNγ production by NK cells.

### Dose-dependent effect of TRAIL on granzyme B production by NK cells

We next addressed whether the mode of action of TRAIL on NK cells was cell-autonomous or cell-extrinsic. To do so, we co-cultured WT and *Trail*$^{-/-}$ splenocytes at different ratios and stimulated them with IL-15 to assess downstream IL-15/IL-15R signaling (Fig 7A).

**Table 1.** REACTOME pathways that are associated with differentially expressed genes in NK cells of WT mice during LCMV infection.

| Reactome ID | Definition | Enrichment | Hypergeometric *P*-value |
|---|---|---|---|
| 194306 | Neurophilin interactions with VEGF and VEGFR | 3/4 | 0.00042 |
| 4641265 | Repression of WNT target genes | 4/9 | 0.00055 |
| 2029485 | Role of phospholipids in phagocytosis | 10/66 | 0.00114 |
| 1236975 | Antigen processing-Cross presentation | 11/90 | 0.00383 |
| 2029481 | FCGR activation | 8/57 | 0.0056 |
| 983170 | Antigen Presentation: Folding, assembly and peptide loading of class I MHC | 6/39 | 0.01009 |
| 392154 | Nitric oxide stimulates guanylate cyclase | 4/17 | 0.00758 |
| 8985947 | Interleukin-9 signaling | 3/10 | 0.01023 |
| 449836 | Other interleukin signaling | 4/19 | 0.01143 |
| 451927 | * Interleukin-2 family signaling | 6/43 | 0.01604 |
| 418346 | Platelet homeostasis | 8/72 | 0.02167 |
| 389357 | * CD28-dependent PI3K/Akt signaling | 4/22 | 0.01926 |
| 139853 | Elevation of cytosolic Ca$^{2+}$ levels | 3/13 | 0.02189 |
| 70895 | Branched-chain amino acid catabolism | 4/23 | 0.02245 |
| 2173782 | Binding and Uptake of Ligands by Scavenger Receptors | 8/79 | 0.03536 |

REACTOME pathways that are associated with differentially expressed genes in NK cells of WT mice during LCMV infection (with a hypergeometric *P*-value ≤ 0.05 and adjusted *P*-value ≤ 0.1). Enrichment indicates the number of differently expressed genes among all genes listed in a specific pathway. Pathways related to PI3K/AKT signaling and IL-2 family signaling are indicated by an asterisk. This table relates to Fig 5.

**Table 2.  REACTOME pathways that are associated with differentially expressed genes in NK cells of *Trail*$^{-/-}$ mice during LCMV infection.**

| Reactome ID | Definition | Enrichment | Hypergeometric *P*-value |
|---|---|---|---|
| 1059683 | Interleukin-6 signaling | 5/11 | 0.00011 |
| 391903 | Eicosanoid ligand-binding receptors | 5/14 | 0.00043 |
| 447115 | Interleukin-12 family signaling | 5/20 | 0.00259 |
| 6799990 | Metal sequestration by antimicrobial proteins | 2/3 | 0.00727 |
| 879518 | Transport of organic anions | 4/15 | 0.00549 |
| 202433 | Generation of second messenger molecules | 5/26 | 0.00855 |
| 425407 | SLC-mediated transmembrane transport | 22/250 | 0.00782 |
| 1442490 | Collagen degradation | 9/71 | 0.00885 |
| 8957275 | Post-translational protein phosphorylation | 13/123 | 0.00885 |
| 382551 | Transport of small molecules | 50/705 | 0.00932 |
| 389948 | PD-1 signaling | 4/19 | 0.01329 |
| 1433557 | Signaling by SCF-KIT | 6/40 | 0.01395 |
| 199418 | * Negative regulation of the PI3K/AKT network | 11/104 | 0.01519 |
| 425397 | Transport of vitamins, nucleosides, and related molecules | 6/41 | 0.01566 |
| 5357801 | Programmed Cell Death | 11/105 | 0.01622 |
| 425393 | Transport of inorganic cations/anions and amino acids/oligopeptides | 11/109 | 0.02088 |
| 1475029 | Reversible hydration of carbon dioxide | 3/12 | 0.01964 |
| 157118 | Signaling by NOTCH | 6/45 | 0.02398 |
| 352230 | Amino acid transport across the plasma membrane | 5/33 | 0.02313 |
| 6811558 | * PI5P, PP2A, and IER3 regulate PI3K/AKT signaling | 10/97 | 0.02353 |

REACTOME pathways that are associated with differentially expressed genes in NK cells of *Trail*$^{-/-}$ mice during LCMV infection (with a hypergeometric *P*-value ≤ 0.05 and adjusted *P*-value ≤ 0.1). Enrichment indicates the number of differently expressed genes among all genes listed in a specific pathway. Pathways related to PI3K/AKT signaling are indicated by an asterisk. This table relates to Fig 5.

We found that S6 phosphorylation and granzyme B expression were directly proportional to the frequency of TRAIL-expressing cells in the culture, irrespectively of the genotype of the analyzed NK cells. Indeed, increased ratios of co-cultured *Trail*$^{-/-}$ splenocytes reduced S6 phosphorylation and granzyme B expression in WT NK cells (Fig 7B-left panel). Conversely, augmentation of the proportion of WT, TRAIL-expressing cells reversed the impaired downstream IL-15/IL-15R signaling of *Trail*$^{-/-}$ NK cells in a dose-dependent fashion (Fig 7B-right panel).

To evaluate the physiological relevance of these findings, we transferred congenic WT (Ly5.1) splenocytes into recipient WT (Ly5.2) and *Trail*$^{-/-}$ (Ly5.2) mice, which were then infected with LCMV (Fig 7C). Similar to the *in vitro* co-culture studies, donor WT NK cells showed decreased S6 phosphorylation when activated in *Trail*$^{-/-}$ compared with WT recipient mice (Fig 7D).

As these results implied a cell-extrinsic contribution of TRAIL to the control of NK cell function, we next examined the role of soluble versus membrane-bound TRAIL for this regulatory mechanism. For

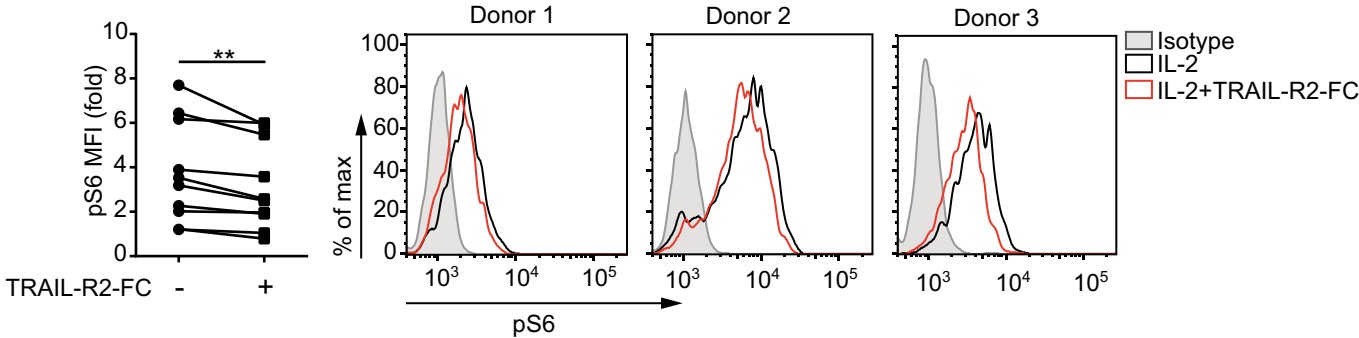

**Figure 6.  TRAIL also regulates the signaling downstream of IL-15 receptor in primary human NK cells.**
Human peripheral blood mononuclear cells were cultured with IL-2 ± human TRAIL-R2-Fc chimeric protein, and S6 phosphorylation was measured in CD56$^{bright}$ NK cells. Data were pooled from three independent experiments (*n* = 9 donors per condition). Values shown were normalized to unstimulated controls. Representative histograms are also shown (right-hand side). Statistical analyses were performed using Wilcoxon matched-pairs signed rank test. **P < 0.01.

this, we used human NK-92 cells that show constitutive expression of TRAIL and its receptors DR4 and DR5, and which are thus inherently endowed with the capacity to engage TRAIL signaling

(Fig EV5K–M). Of note, NK-92 cells were found to behave similarly to primary murine and human NK cells, since TRAIL blockade also repressed the signaling downstream of IL-15 receptor in this cell line

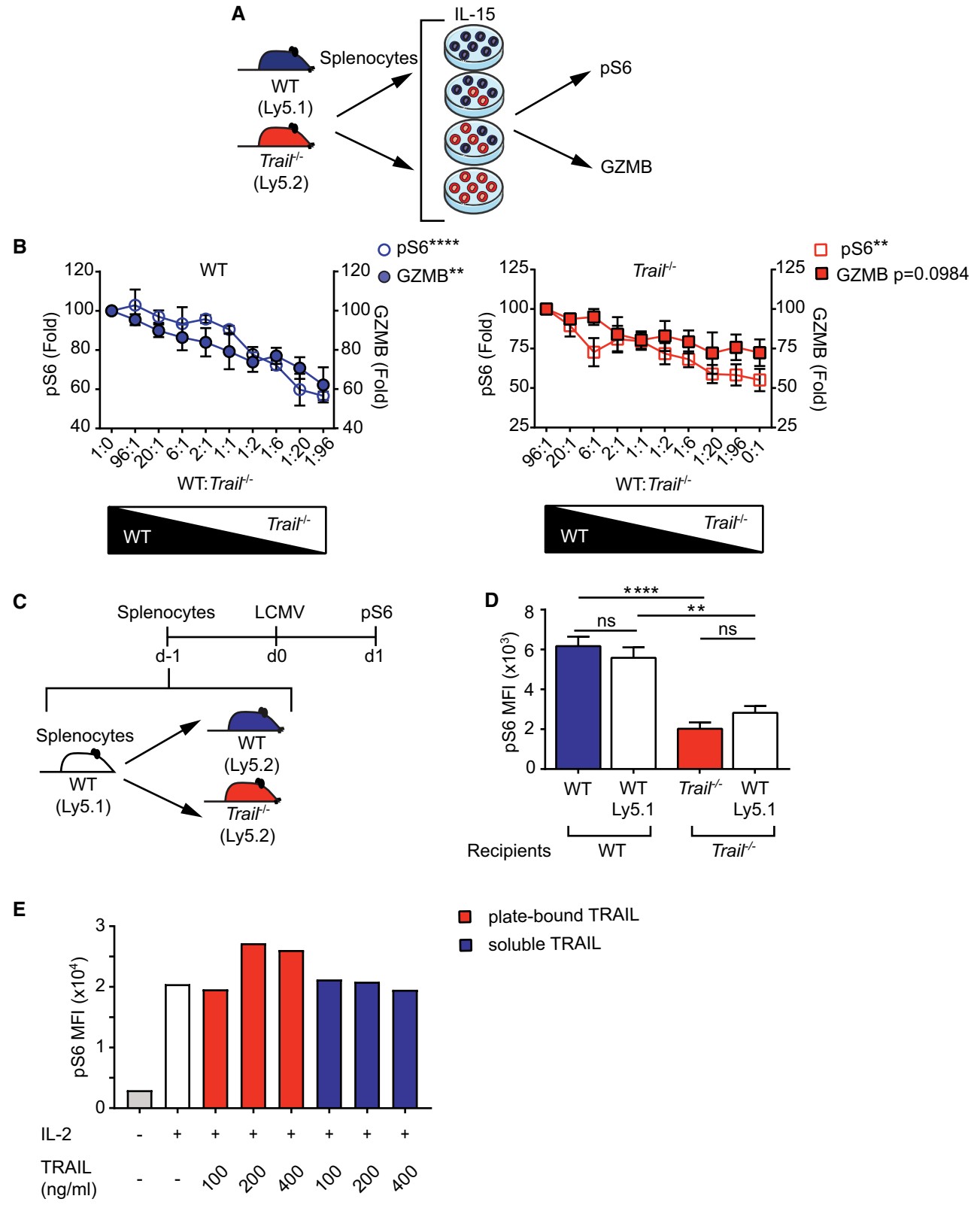

**Figure 7.**

◄

**Figure 7.  Dose-dependent effect of TRAIL on NK cell function.**

A, B    Experimental setup of co-culture assays (A). WT and $Trail^{-/-}$ splenocytes were co-cultured and at the indicated ratios and stimulated with IL-15 to assess NK cell
        pS6 and GZMB expression (B). Data were normalized to mean fluorescence intensity (MFI) levels of cultures containing only WT (left panel) or $Trail^{-/-}$ (right panel)
        cells. Data shown are mean ± SEM and are representative of three independent experiments ($n$ = 6 for GZMB and $n$ = 3 for S6). Statistical analyses were
        performed using one-way ANOVA. **$P$ < 0.01; ****$P$ < 0.0001.
C, D    Experimental setup of adoptive transfers (C). WT (Ly5.1) donor splenocytes ($10^7$) were transferred into WT (Ly5.2) or $Trail^{-/-}$ (Ly5.2) recipient mice that were then
        infected with LCMV. Expression of pS6 was assessed after 24 h on donor and host NK cells (D). Data shown are mean ± SEM and are representative of two
        independent experiments ($n$ = 4). Statistical analyses were performed using one-way ANOVA with Tukey post-test. ns, non-significant; **$P$ < 0.01; ****$P$ < 0.0001.
E       NK-92 cells were cultured at various concentrations of plate-bound or soluble recombinant TRAIL followed by stimulation with IL-2. Phosphorylation of S6 was
        measured by flow cytometry. Data are representative of two independent experiments ($n$ = 1 per condition).

(Fig EV5N and O). Remarkably, addition of plate-coated—but not soluble—recombinant TRAIL promoted, in a dose-dependent manner, IL-15 signaling in NK-92 cells (Fig 7E). Therefore, cell-extrinsic membrane-bound TRAIL activates TRAIL signaling in NK cells during inflammation to dose-dependently regulate granzyme B production.

Collectively, our findings uncover a non-canonical role of TRAIL signaling for the modulation of essential NK cell functions, which restrains T-cell response and thereby determines the course of LCMV infection (Appendix Fig S1).

## Discussion

In this study, we identified a novel, non-apoptotic role of TRAIL as a modulator of cytokine production and cytotoxic granule content in NK cells, during viral infection. Our data indicate that $Trail$-deficient NK cells have reduced granzyme B expression that is associated with impaired NK cell-mediated restriction of T-cell response. This resulted in better virus control and reduced liver pathology in LCMV WE-infected $Trail^{-/-}$ animals.

A previous study reported no alteration in the primary CD8$^+$ T-cell response of $Trail^{-/-}$ mice infected with the Armstrong (ARM) strain of LCMV [38]. In agreement with these findings, we found that, compared to WT control recipients, $Trail^{-/-}$ mice only showed an increased expansion of donor $Trail$-competent transgenic P14 cells when they were infected with LCMV-WE, but not with LCMV-ARM (Appendix Fig S2). These $Trail$-dependent differences in the T-cell response induced by LCMV-WE compared to LCMV-ARM are likely due to distinct virus-specific CD8$^+$ T-cell expansion magnitude or kinetics, which are directly related to different kinetics of virus replication between LCMV-WE and LCMV-ARM strains [39]. Thus, it is conceivable that the $Trail$-controlled modulation of NK cell activity may differently regulate the virus-specific CD8$^+$ T-cell response in dependence of its extent or kinetics.

NK cells directly interact with T cells via the engagement of activating and inhibitory ligands whose integration determines overall NK cell function [40–42]. Although granzyme B production in $Trail^{-/-}$ NK cells was markedly altered, there were no substantial differences in NK cell receptor expression in $Trail^{-/-}$ versus WT NK cells. Of the NK cell receptors analyzed, only Ly49H was affected in its expression, with decreased levels on $Trail^{-/-}$ NK cells. Ly49H is an activating NK receptor that recognizes the m157 glycoprotein of murine cytomegalovirus (MCMV) on infected cells. Engagement of Ly49H triggers activation, robust expansion and differentiation of NK cells into a long-lived memory subset [43,44]. While the function of this receptor during LCMV infection is unknown, Ly49H-positive NK cells from LCMV-infected mice—but also from naïve animals—produce more IFNγ than their Ly49H-negative counterparts when stimulated $ex$ $vivo$ by pro-inflammatory cytokines [45]. In contrast, our results indicate that whereas $Trail^{-/-}$ NK cells exhibit lower Ly49H expression than WT cells during LCMV-WE infection, they show increased IFNγ production. Together, these findings suggest that TRAIL has only minor effects—if any—on NK cell development and that it likely does not alter the ability of NK cells to recognize activated T cells.

Our data establish that TRAIL positively modulates IL-15 signaling-induced granzyme B production in NK cells. A previous study reported no change in the cytotoxicity of NK cells from MCMV-infected $Tnfrsf10b/TRAIL-R2^{-/-}$ mice [46]. These different findings might be explained by the fact that upon MCMV infection $Il15ra$- or $Il15$-deficient NK cells are still capable to secrete cytokines and exert cytotoxicity. Indeed, NK cell effector functions during MCMV infection chiefly rely on the m157 viral glycoprotein-induced activation of Ly49H receptor on NK cells [47].

IL-15 not only shapes the function of NK cells. It is also of central relevance for the development, homeostasis, and proliferation of these cells [48], which is induced downstream of IL-15/IL-15R signaling by phosphorylation of the transcription factor STAT5 [49]. In spite of a reduced ability to engage IL-15/IL-15R signaling upon activation, NK cells in $Trail^{-/-}$ mice were found to develop similarly as in WT animals, with comparable numbers at steady state and upon LCMV infection. As mTOR activation requires higher concentration of IL-15 than STAT5 phosphorylation [35], we therefore conclude that TRAIL only regulates IL-15/IL-15R signaling associated with PI3K-mTOR pathway.

Our results indicate that DR5 is upregulated on NK cells during LCMV infection, but not on virus-specific CD8$^+$ T cells. Although this strongly suggests that in our model NK cells control T cells independently of TRAIL pro-apoptotic function, we cannot formally exclude that TRAIL-induced cell death may be engaged in other cells $in$ $vivo$, which might subsequently affect NK cell function.

Besides the well-characterized induction of caspase-dependent apoptosis, TRAIL/TRAIL-R signaling may also activate non-canonical pro-survival pathways through activation of NF-κB, ERK1/2, and PI3K/AKT pathways [50,51]. Furthermore, TRAIL treatment induces rapid phosphorylation of Akt and mTOR in TRAIL-resistant cancer cell lines [52,53]. While this non-canonical TRAIL-PI3K-AKT signaling has been described in transformed cells, its precise underlying molecular mechanisms are still unclear [54]. Here, we provide evidence that a similar mechanistic link between TRAIL and the PI3K-mTOR pathway exists in primary NK cells. Hence, TRAIL appears to enhance IL-15/IL-15R signaling by synergistic induction of AKT phosphorylation and subsequent mTOR activation.

Our finding of a negative contribution of TRAIL signaling to IFNγ secretion by NK cells relates to a previous report indicating increased serum IFNγ and IL-12 in MCMV-infected *TRAIL-R2*$^{-/-}$ mice. However, *TRAIL-R2*$^{-/-}$ mice did not show higher frequency of IFNγ$^{+}$ NK cells in this infection model. Rather, enhanced IL-12 producing by *TRAIL-R2*$^{-/-}$ DCs likely induced increased numbers of IFNγ-secreting NK cell upon MCMV infection [46].

IFNγ signaling is critical to restrict the spread of various LCMV strains at the onset of infection and therefore promote the virus-specific CD8$^{+}$ T-cell response and later pathogen clearance [55–58]. Yet, our data from NK cell depletion experiments indicate that the higher levels of IFNγ that are produced by NK cells in LCMV-infected *Trail*$^{-/-}$ mice do not account for the increased CD8$^{+}$ T-cell response in these animals. This conclusion is further supported by the findings from the *in vivo* NK cell-mediated killing assay, suggesting that reduced NK cell cytotoxicity, rather than increased IFNγ production, regulates the expansion of virus-specific CD8$^{+}$ T cells in infected *Trail*$^{-/-}$ mice. Although IFNγ can affect donor T-cell survival, motility, and migration [59–61], this effect is unlikely to happen within the 6-h window of this *in vivo* cytotoxicity assay, as also suggested by a previous study using as a readout transgenic CD8$^{+}$ T cells and antibody-mediated IFNγ blockade [59–61].

We found that crosslinking of the activating NK cell receptor NK1.1 leads to enhanced IFNγ production in *Trail*$^{-/-}$ compared with WT NK cells, implying an inhibitory effect of the TRAIL/ TRAIL-R pathway on the signaling downstream of NK1.1. Activation of NK1.1 engages a CARMA1/BCL10/MALT1 complex that induces JNK and MAPK phosphorylation and promotes canonical NF-κB pathway. While this pathway was shown to be largely dispensable for NK cell-mediated cytotoxicity, it specifically controls the production of cytokines including TNF and IFNγ [62]. Absence of TRAIL-R is associated with prolonged NF-κB pathway activation in stimulated DC and macrophages, possibly via regulation of IκB-α degradation or stability [46]. Therefore, the TRAIL/TRAIL-R axis might similarly regulate activating NK cell receptor-dependent cytokine production via modulation of NF-κB signaling.

NK cell activation by cytokines or engagement of NK cell receptors can promote cytokine secretion or cytotoxicity either conjointly or independently [48,63]. Our results indicate that TRAIL constrains the ability of activated NK cells to secrete cytokine while enhancing their cytotoxic potential.

NK cells may show divergent roles during infection and subsequent pathophysiology, in a microbe- or organ-specific manner [64]. NK cell-derived cytokines can promote infection-induced immunopathology [65–68]. Yet, IFNγ produced by NK cells during infection can be also beneficial for recovery—while at the same time NK cytotoxic activity is detrimental [69]. The dual regulatory role of TRAIL in NK cells may have evolved in such context to fine-tune NK effector function and balance protective immunity versus immunopathology.

In conclusion, these findings reveal an unprecedented and unexpected contribution of TRAIL to the control of NK cell function, independently of its pro-apoptotic role. Furthermore, they also show the relevance of this novel regulatory mechanism for the modulation of CD8$^{+}$ T-cell response and subsequent virus control. Our results warrant further investigation on whether manipulation of TRAIL signaling may be exploited for NK cell-based immunotherapy.

# Materials and Methods

## Mice

C57BL/6J mice were purchased from Jackson Laboratories (Bar Harbor, ME, USA) and subsequently bred in-house. *Tnfsf10/Trail*$^{-/-}$ [70] (referred to as *Trail*$^{-/-}$ mice), *Prf1*$^{-/-}$ [71], congenic B6.SJL-Ptprc$^{a}$Pepc$^{b}$/BoyJ (Ly5.1), and Tg(TcrLCMV)327Sdz;*Rag1*$^{tm1Mom}$; B6.SJL-Ptprc$^{a}$Pepc$^{b}$/BoyJ (P14) [72] mice were housed and bred in specific pathogen-free facilities. Annette Oxenius (ETH Zurich, Switzerland) kindly provided us with spleens from B6.129S7-*Ifnar1*$^{tm1Agt}$;Tg (TcrLCMV)327Sdz;B6.PL-$^{Thy1a/CyJ}$ (Thy1.1$^{+}$ *Ifnar1*$^{-/-}$ P14) mice [73]. *Trail*$^{-/-}$ mice were backcrossed at least ten times onto a C57BL/6 background, and all mice were maintained on a C57BL/6 genetic background. For all experiments, non-randomized groups of 8- to 12-week-old females or males were used. Animal experiments were carried out in compliance with the ARRIVE reporting guidelines. All experiments were performed in accordance with Swiss Federal regulations and were approved by the Cantonal Veterinary Office of Bern, Switzerland.

## Antibodies and reagents

All fluorescent-labeled antibodies used in this study are indicated in Appendix Table S1. LCMV-specific peptides GP$_{33–41}$ (KAVYNFATC), GP$_{61–80}$ (GLKGPDIYKGVYQFKSVEFD), and NP$_{396–404}$ (FQPQNGQFI) were purchased from Eurogentec (Lüttich, Belgium). Cell proliferation dye eFluor 670 was obtained from eBioscience (Santa Clara, CA, USA).

## Virus and virus titration

LCMV strain WE was obtained from Stefan Freigang (Institute of Pathology, University of Bern, Switzerland) and was propagated at a low multiplicity of infection on L929 fibroblast cells. Virus titers were measured using a plaque-forming assay, as previously described [72]. Mice were injected intravenously (i.v.) with 10$^{5}$ plaque-forming units (pfu) LCMV strain WE, a dose range shown to induce liver immunopathology [20]. In the Fig 1C, group size is as follows: for each group of mice, virus titers in the spleen: day 1, $n = 9$; day 2, $n = 9$; day 3, $n = 3$; day 5, $n = 3$; day 7, $n = 3$; day 8, $n = 9$; day 10, $n = 11$; day 12, $n = 6$; virus titers in the liver: day 1, $n = 3$; day 2, $n = 3$; day 3, $n = 3$; day 5, $n = 3$; day 8, $n = 6$; day 10, $n = 8$; day 12, $n = 3$.

## NK cell depletions

NK cells were depleted either with 200 μg of anti-NK1.1 (clone PK136, BioXcell, New Hampshire, USA), as performed in a previous study [26], or 20 μl of anti-asialo GM1 antibody, as recommended by the manufacturer (Wako Pure Chemical Industries, Virginia, USA). Depletions were performed by intraperitoneal (i.p.) injection of the respective depleting antibodies 3 and 1 days before LCMV infection. Saline (PBS) was injected as a negative control. The efficacy of NK cell depletion was verified.

## Adoptive cell transfers

CD8$^{+}$ T cells were purified from naïve P14 spleens by immune-magnetic negative selection (Miltenyi Biotec, Bergisch Gladbach,

Germany), and $10^6$ cells were transferred per recipient mouse 48 h after LCMV infection. P14 cells were analyzed for expansion and activation 2 and 5 days after transfer, respectively.

Alternatively, $8 \times 10^6$ wild-type (Ly5.1) splenocytes were transferred per wild-type or $Trail^{-/-}$ (Ly5.2) recipient mouse 16 h before infection and both donor and recipient NK cells were analyzed by flow cytometry after LCMV infection.

### Quantification of alanine aminotransferase and cytokine levels in serum

Alanine aminotransferase (ALT) concentration in the serum was measured at the Department of Clinical Chemistry of the Inselspital/ Bern University Hospital using a Roche Modular P800 Analyzer (Roche Diagnostics, Rotkreuz, Switzerland).

Serum cytokine levels were measured using a mouse IFNγ ELISA set kit (BD Biosciences, San Jose, CA, USA) or by flow cytometry using a bead-based multiplex assay (LEGENDplex, BioLegend, San Diego, CA, USA). Type I IFN activity was measured using an L-929 cell line transfected with an interferon-sensitive luciferase construct [74].

### Isolation of spleen and liver lymphocytes

Spleens were homogenized to single-splenocyte suspensions using a 70 μm cell strainer (Falcon Technologies BD, NY, USA). For livers, lymphocytes were isolated from single-cell suspensions by Percoll gradient centrifugation ($800 \times g$, 15 min).

### Flow cytometry

For intracellular cytokine staining, lymphocyte preparations were first incubated for 5 h at 37°C with 20 μg/ml Brefeldin A (Sigma-Aldrich, St. Louis, MO, USA) (for NK cells), or with 20 μg/ml Brefeldin A and $10^{-7}$ M of a particular LCMV-specific peptide (for T cells). After incubation, cells were stained for surface markers for 20 min. Cells were then washed, fixed, and permeabilized using BD Bioscience Cytofix/Cytoperm solution, followed by intracellular staining with anti-IFNγ and anti-TNF antibodies.

For intracellular staining of phosphorylated proteins, cells were fixed using paraformaldehyde 4% w/v in PBS and methanol 90% v/v in water.

Data were acquired on a LSRII flow cytometer (BD Bioscience) and analyzed using a FlowJo software (Tree Star Inc., Ashland, Oregon, USA).

### Isolation and RNA sequencing of NK cells

Single-cell suspensions were prepared from spleens isolated from naive ($n = 3$ per group) or from LCMV-WE-infected ($n = 4$ per group) WT or $Trail^{-/-}$ mice, 1 day postinfection. NK cells (defined as NK1.1$^+$ and CD3$^-$ cells) were sort-purified by flow cytometry and resuspended in TRI-reagent (Sigma-Aldrich). RNA was isolated according to the manufacturer's instructions, and RNA concentration and integrity were assessed using a Bioanalyzer 2100 (Agilent, Santa Clara, CA).

Barcoded stranded mRNA sequencing libraries were prepared from high-quality total RNA samples (~10 ng/sample) using combination of the NEBNext Poly(A) mRNA Magnetic Isolation Module (NEB #E7490, Ipswich, MA, USA) to enrich the samples for polyadenylated RNA transcripts and the Ultra II Directional RNA Library Prep Kit (NEB #E7760). Obtained libraries that passed the quality check step were pooled in equimolar amounts, and 1.8 pM solution of this pool was loaded on the Illumina sequencer NextSeq 500 and sequenced uni-directionally, generating ~500 million reads, each 85 bases long. Library preparation and sequencing was performed at the EMBL Genomics Core Facilities (GeneCore, Heidelberg, Germany).

### Computational analysis of RNA sequencing data

RNA sequencing (RNA-seq) data processing was performed on the SevenBridges platform [75]. Between 39.9 and 47.6 million reads were obtained per sample. Read quality was assessed using FastQC [76], and STAR [77] was applied to align the reads to the reference genome (Ensembl m38, build 93) [78]. We then used HTSeq-count [79] to count the number of reads per gene. Differential expression analysis was performed with DESeq2 [80] to identify genes for which the relative frequency of transcripts differed in NK cells upon infection with LCMV-WE (i.e., transcripts were compared before and after infection and selected for adjusted *P*-value < 0.01 and absolute log2 fold change ≥ 2). Samples of WT and *Trail*-deficient populations were then compared separately to extract two lists of differentially expressed genes.

### Pathway analysis

BioInfoMiner [81] was applied for the functional interpretation of the differently expressed genes, using the Biological Process domain of Gene Ontology (GO). BioInfoMiner is an automated tool embedded on the SevenBridges platform that was created for the translational analysis of genomic data. It takes into account the topological organization of terms, targeting to correct the ontological annotation, while it adopts a non-parametric statistical correction during the enrichment analysis in order to ensure the promotion of non-trivial, system-level terms. For the analysis, two lists of significant pathways were extracted, one for each comparison. To disclose differences between these pathway lists, the parent–child relations of Gene Ontology graph were exploited. The difference between a set of terms N to a set M could be defined as those terms which belong to N, so that neither their descendants nor themselves are included in M. Using this approach, unique terms were revealed for each pathway list. In order to remove potential redundancy, terms whose descendants were also significant were filtered out. Thereby, starting from the functional interpretation of differentially expressed genes, the analysis ended to disclose the most specific, uniquely associated biological processes found to be affected during infection for each genotype.

Alternatively, REACTOME [82] was used instead of Gene Ontology as a reference database, with BioInfoMiner as an analysis tool.

Data from the RNA-seq analysis with detailed gene and pathway lists are provided in the source data files related to Figs EV2 and EV3.

### NK cell cytotoxicity assays

For the *ex vivo* NK cell cytotoxicity assays, total splenocytes were isolated on day 5 post-LCMV-WE infection and incubated for 5 h and

at different effector/target ratios with eFluor 670-stained YAC-1 cells. The number of effector cells was adjusted based on NK cell frequencies among splenocytes. Live/dead discrimination of YAC-1 cells was performed using annexin V (BioLegend) and DAPI (BioLegend).

For the *in vivo* NK cell cytotoxicity assay, $5 \times 10^7$ splenocytes from *Ifnar1*$^{-/-}$ P14 mice were transferred into NK cell-depleted WT recipient mice that were then infected with LCMV. Four days after infection, splenic CD8$^+$ T cells were MACS-purified and transferred into groups of day 4 LCMV-infected recipient mice. Frequencies of *Ifnar1*$^{-/-}$ P14 cells in the spleen were measured 6 h after transfer.

### Quantitative PCR analysis

Total RNA was isolated from mouse spleen and liver tissue using TRI-reagent (Sigma-Aldrich) according to the manufacturer's instructions. RNA was reverse-transcribed into cDNA using M-MLV Reverse Transcriptase (Promega, Fitchburg, WI, USA). FastStart SYBR Green Master (Roche, Basel, Switzerland) and commercial primers (Qiagen, Venlo, Netherlands) were used to detect *Trail*, *Il15*, and *Gapdh* transcript levels. All PCR products were run and analyzed on a StepOnePlus Real-Time PCR System (Life Technologies, Carlsbad, CA, USA). Expression levels of the tested genes were normalized to *Gapdh* mRNA, or fold induction was calculated using the $2^{-\Delta\Delta CT}$ method [83].

### *In vitro* cytokine stimulation of NK cells and TRAIL signaling inhibition studies

Splenocytes were isolated from naïve mice and cultured with recombinant murine IL-15 (100 ng/ml) (PeproTech, NJ, USA). For PI3K pathway inhibition, 1 μM wortmannin and 50 μM LY294002 were added 1 h prior to activation with IL-15. TRAIL signaling was blocked using 3 μg/ml of mouse TRAIL-R2-Fc chimeric protein (Enzo Life Sciences, NY, USA) added 1 h prior to activation with IL-15. Kinase phosphorylation was assessed 1 h and granzyme B expression 20 h after the addition of IL-15.

Whole blood samples from health donors were obtained from Interregional Blood Transfusion SCR Ltd, Bern, Switzerland, under the signed consent of the donors and in agreement with the local legislation. Human peripheral blood mononuclear cells were isolated by Ficoll gradient centrifugation and were stimulated for 1 h at 37°C with 1,000 UI/ml of recombinant human IL-2, as previously described [35]. TRAIL signaling was blocked using 3 μg/ml of human TRAIL-R2-Fc chimeric protein (Enzo Life Sciences).

### Co-culture of wild-type and *Trail*$^{-/-}$ splenocytes

Splenocytes isolated from naïve wild-type (Ly5.1) and *Trail*$^{-/-}$ (Ly5.2) mice were co-cultured at different ratios and stimulated with IL-15 (100 ng/ml) for 1 h to assess kinase phosphorylation or for 20 h to evaluate granzyme B expression.

### *In vitro* experiments using NK-92 cells

Human NK-92 cells were initially obtained from ATCC and kindly provided to us by Eva Szegezdi (NUI, Galway, Ireland). Cell lines were tested negative for mycoplasma. NK92 cells were maintained in medium supplemented with 100 UI/ml of IL-2. To assess the mechanisms underlying the immunoregulatory effect of TRAIL signaling on NK cells, NK-92 cells were first washed and cultured for 16 h in medium not supplemented with IL-2. NK-92 cells were next incubated at 37°C with different concentrations of plate-coated or soluble human TRAIL (BioLegend) for 30 min, and IL-2 (2,000 UI/ml) was then added to these cultures for one or 20–24 more hours before flow cytometry analysis of S6 phosphorylation or GZMB expression, respectively.

### Statistical analysis

Sample size for *in vivo* studies was estimated by power analysis and adjusted for β = 0.2, with the assumption that differences between the groups were 1.5- to 2-fold. Statistical tests were selected based on the variation in each data group and on whether multiple comparisons were performed. Groups with similar variance were compared using parametric tests; groups with significantly different variations were analyzed using non-parametric tests. For datasets including several time points, tests were selected based on the type of variance present in the majority of the time points. Statistical tests are two-tailed and indicated in the figure legends. All statistical evaluations were performed using GraphPad Prism v.7.0b for Mac or v.6.03 for Windows (GraphPad Software, La Jolla, CA, USA). Unless specified, only statistically significant differences are indicated in the figures. For all statistical analyses: $*P < 0.05$; $**P < 0.01$; $***P < 0.001$; $****P < 0.0001$.

## Data availability

The datasets produced in this study are available in the following database: RNA-Seq data: ArrayExpress E-MTAB-7562 (https://www.ebi.ac.uk/arrayexpress/experiments/E-MTAB-7562/).

**Expanded View** for this article is available online.

### Acknowledgements

We thank Regula Stuber for her excellent technical support. We are grateful to Annette Oxenius (ETH Zurich, Switzerland) and Eva Szegezdi (NUI, Galway, Ireland) for providing us with reagents and to Vladimir Benes (GeneCore, Heidelberg, Germany) for advice on RNA isolation methods. We also like to extend our gratitude toward Haifeng C. Xu, Werner Held, Stefan Freigang, Daniel L. Popkin, Antoine Marçais, Christian M. Schürch, Mario Noti, and Lukas F. Mager for advice or critical comments. This work was supported by grants from the Swiss National Science Foundation (310030_138188 and 314730_163086), the "Vontobel Stiftung", the "Olga Mayenfisch Stiftung", the "Kurt und Senta Herrmann-Stiftung", a generous donor advised by CARIGEST SA, the Bern University Research Foundation (all to P.K.). This project has also received funding from the European Union Seventh Framework Program (FP7) under grant agreement No PCIG12-GA-2012-334081 (X-talk) (to P.K.) and the European Union's Horizon 2020 research and innovation program under the Marie Skłodowska-Curie grant agreement No 777995 (DISCOVER) (to P.K. and A.C.).

### Author contributions

LCA conceived and performed experiments and wrote the manuscript. MDB, NK, and CL performed experiments. TK and AC performed computational analysis. RS provided key reagents. NC provided expertise and feedback. PK conceived experiments, wrote the manuscript, and secured funding.

## Conflict of interest

The authors declare that they have no conflict of interest.

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
