## [Review Process File · EMBO Reports]

Non-apoptotic TRAIL function modulates NK cell activity during viral infection

Ludmila Cardoso Alves, Michael D. Berger, Thodoris Koutsandreas, Nick Kirschke, Christoph Lauer, Roman Spörri, Aristotelis Chatziioannou, Nadia Corazza, Philippe Krebs

Review timeline:

Submission date:	3 July 2019
Editorial Decision:	21 August 2019
Revision received:	24 September 2019
Editorial Decision:	8 October 2019
Revision received:	16 October 2019
Accepted:	18 October 2019

Editor: Achim Breiling

Transaction Report:

Please note that the manuscript was previously reviewed at another journal. As EMBO Press has a transfer agreement (including the identities of the referees) with that journal, revision was invited based on the reports from that previous external submission upon request by the authors.

REFeree REPORTS

Reviewer #1:

In this manuscript, Alves et al study the effect of TRAIL-deficiency on control of LCMV infection and discover enhanced early virus-specific CD8+ T cell responses without major effects on virus titers until later. Anti-NK1.1 depletion enhanced CD8+ responses and abrogated TRAIL-dependent effects on CD8+ cells but no increases in NK cell numbers were seen. TRAIL deficiency was associated with increased IFN γ production by NK cells that the authors concluded was not related to the augmented CD8 responses in TRAIL-deficient mice. By contrast, they provide evidence for decreased GZB expression in NK cells in TRAIL-deficient mice that they suggest is the basis for enhanced CD8 responses. In vitro experiments suggest lower IL15 stimulated responses in TRAIL-def NK cells that the authors attribute to lower GZB production. However, they then show that WT NK cells show decreased NK cell responses when adoptively transferred into TRAIL-deficient recipients, suggesting NK cell extrinsic effects.

The most challenging aspect of this manuscript are the multiple, apparently contradictory results that confound interpretations and attempts to come up with a unifying explanation for the initial findings that there are enhanced virus-specific CD8+ T cell responses in LCMV infections of TRAIL-deficient mice. For example, the authors conclude that IFN γ production by NK cells is not relevant to the enhanced CD8 response but they still study this aspect of NK cell function later on in the manuscript. In addition, a number of conclusions are reached that are not directly supported by the experiments presented. For example, the authors conclude that GZB production is relevant to the phenotype but do not directly test if NK cell cytotoxicity of CD8 T cells is actually operational in their studies. They show many experiments suggesting that TRAIL affects

IL15 signaling in NK cells, but other data suggest that the effects in vivo are due to NK cell extrinsic effects.

That TRAIL appears to affect IL15 signaling in NK cells is quite interesting and apparently novel but there are no experiments to address how this occurs.

Minor Comments:

Fig 1C. Not clear why two asterisks at day 8 and one at day 10 when there are minimal effects on titers.

Fig 2C. Not clear what cells are being studied.

Line 126. Premature to conclude that TRAIL is affecting NK cell activity.

Fig 2D. Should be replotted to show anti-NK1.1 next to untreated for WT, and for TRAILdef.

Reviewer #2:

This manuscript describes a non-canonical role for TRAIL in determining NK-cell function during viral infection. They conclude that cell-extrinsic TRAIL engages TRAIL-R on NK cells, supports IL-15 signaling and promotion of granzyme B while reducing IFN γ . Consequently, NK cells suppress CD8 T cells and alter viral pathogenesis. The experiments appear sound and represent important findings that will be intriguing to the scientific community. However, several issues reduce confidence in the interpretation of the data and potentially undermine the mechanistic conclusions. What is the evidence that the phenomenon is independent of pro-apoptotic functions of TRAIL? Apoptosis is only examined in Figure 5.S2, where sTRAIL does not trigger apoptosis in NK cells. Yet, the authors don't delineate which specific cell express TRAIL or TRAIL-R, thereby precluding determination of apoptosis in the relevant lineage. The role of membrane-bound TRAIL rather than soluble TRAIL is never formally tested. The non-apoptotic conclusion must be supported by data on the ability/inability of TRAIL or TRAIL-R engagement by soluble or membranous ligands to induce NK cell apoptosis. Moreover, the potential for TRAIL-mediated apoptosis of accessory cells that subsequently influences NK cell function should be discussed.

The current presentation creates unnecessary confusion about the directionality and cell-intrinsic nature of events. None of the data support the necessity or sufficiency of TRAIL-R engagement on NK cells driving these effects, while Fig 5.S2 casts some doubt on this hypothesis. The one experiment clearly performed with isolated NK cells (Fig 5.S3) suggests that Trail^{-/-} NK cells are defective in response to NK1.1 stimulation, which stands in contrast to the conclusion that this is mediated via TRAIL-R engagement. Analysis of NK-cell expression of IFN- γ and granzyme B in Figure 7C/D, or in vitro experiments comparing effect of sTRAIL/mTRAIL/TRAIL-R stimulation on these functions of NK cells, would boost confidence in the conclusions. The conclusion that lower GzmB expression results in reduced suppression of CD8 T cells is tenuous. Do the authors have any data that support a role for GzmB in suppression? The authors should test that TRAIL-deficient, gzmB-low NK cells exhibit reduced capacity to suppress T cells (in vitro perhaps). Ideally, the authors would also show that suppressive function can be rescued by restoring gzmB expression in Trail^{-/-} NK cells, or mimic effect by knocking-down GzmB in wild-type NK. Given the difficulty of the latter experiments, a thorough discussion of the limits of the data should be provided at a minimum.

The role of enhanced IFN- γ is incompletely addressed. Figure 2B (24 hours), Fig 2-S2 (36 h), and Fig 3 (24-48h) show early IFN- γ expression, but enhancement of IFN- γ at later time points is not examined. The possibility that continued elevations in IFN- γ levels can contribute undermines the conclusions of Fig 3F that is purported to show effects on CD8 T cells are independent of early IFN-g. A better experiment would be administration of anti-IFN- γ blocking antibodies during initial days of infection to reverse (or not) enhanced CD8 T cell responses in Trail^{-/-}.

Minor Comments:

1. Were proper control antibodies (for anti-NK1.1 or asialoGM1) administered to the non-depleted group? If not, then additional experiments should be performed with proper controls. How were the doses of anti-NK1.1 and anti-asialoGM1 selected? Was NK cell depletion verified? Please change text of Methods, Results, and/or Figure legends to denote answers to these inquiries.
2. The authors assert that the partial phenotype of Trail-deficiency, relative to NKdepletion, reflects overlapping roles of NK cells and Trail. This could just as likely represent a separate effect of NK cell depletion (like enhanced CD4) that masks or overrides the Trail effect. Detailed discussion of potential interpretations is warranted.
3. More precision and consistency are needed in Results/Methods/Legends regarding whether total splenocyte or isolated NK cells are assayed.
4. In Fig 1C, titers are inappropriately presented and statistics incorrectly applied for day 12. An undetectable virus load cannot be accurately displayed as 0, but should instead be assigned a less-than value just below the L.O.D. Your statistics should be recalculated with these values. Also, not clear if the "***" in spleen viral load plot is incorrectly labeling day 8 when it should be day 7.
5. In Figure 2C, legend should reflect whether these histograms are gated on NK cells (as described in results). Either the results or legend need to be edited.
6. The *Tnfsf10*^{-/-} mice were originally made in 129 stem cells, such that the low Ly49H expression could reflect incomplete backcrossing along chromosome 6. The authors should comment on this and any other potential effects of 129 gene carryover that might impact their measurements. This possibility may negate many of the arguments discussed in the 2nd paragraph of the discussion.
7. In Figure 4, do E:T reflect "splenocyte" to target or "calculated NK cell within splenocyte" to target ratios?

Reviewer #3:

This manuscript presents data indicating that, following LCMV infection, TRAILdeficient mice display 1) enhanced LCMV GP33-41-specific CD8⁺ T cell response, 2) faster pathogen clearance, 3) lower ALT/higher IFN γ levels in sera, and 4) higher IFN γ /pAKT/pS6/lower granzyme B levels in NK cells, compared to WT mice. Based on NK cell depletion study, the authors argue that some of these effects are mediated by NK cells; for instance, the enhanced T cell response in TRAIL-deficient mice is associated with low cytotoxic activity of NK cells resulting from low granzyme B expression. Importantly, the authors provide evidence that TRAIL-TRAILR engagement on NK cells can augment IL-15-induced granzyme B expression by increasing AKT/S6 phosphorylation, which might explain reduced LCMV-specific T cell response and slower pathogen clearance in WT mice.

This manuscript is very interesting and has some important novelties. However, the following points should be addressed to sustain the claims and improve the quality of the manuscript.

1. Based on the available information (Reference #57), TRAIL-deficient mice used in this work were initially produced on non-B6 background, then backcrossed only five times to B6 mice. The manuscript indicates "All mice were maintained on a C57BL/6 genetic background", which needs to be further clarified. The genetic background of TRAILdeficient mice and WT mice should be extensively matched to exclude the possible non-TRAIL-related effects.
2. A previous study (Badovinac et al, 2006, not cited) reported that there was no difference in LCMV-specific CD8⁺ T cell responses, including GP33-41-specific response, between TRAIL-deficient mice (backcrossed 10 times to B6) and WT B6 mice. This discrepancy should be discussed.
3. Abstract indicates "Our study reveals...immunoregulatory role of TRAIL signaling on NK cells for the granzyme B-dependent elimination of antiviral T cells", but no clear evidence to support this claim was presented in the manuscript. Is there any evidence

that TRAIL-deficient NK and WT NK cells kill directly and differentially LCMV-specific T cells? If not, this could be misleading and should be rephrased. Additionally, possible direct interactions (e.g., via activating receptors) between NK and T cells should be discussed.

4. Examination of granzyme B expression, pAKT/pS6 and NK1.1-induced IFN γ production following adoptive transfer (eg, Fig. 7D; and transfer of TRAIL-deficient spleen cells into WT mice) may yield informative data. NK1.1-crosslinking experiments were performed on MACS-purified DX5⁺ cells (Fig. 5 Suppl. 3D), but more accurate data might be produced with whole splenocytes with less complications.

Authors' Reponse

**Comments of Reviewer #1:**56 **Reviewer #1 / Major comments:**

*1. The most challenging aspect of this manuscript are the multiple, apparently contradictory*
*results that confound interpretations and attempts to come up with a unifying explanation for*
*the initial findings that there are enhanced virus-specific CD8⁺ T cell responses in LCMV*
*infections of TRAIL-deficient mice.*

We agree with Reviewer #1 that the first submitted version of our manuscript may have
appeared confusing to the reader, in particular concerning the mechanisms – cell-autonomous
versus cell-extrinsic mechanisms – by which TRAIL signaling affects NK cell function. Some
information was missing, which prevented proper understanding of our results. This has also
been mentioned by the other Reviewers (e.g. minor comment #2 of Reviewer #1; major
comment #3 and minor comments #3 and #5 of Reviewer #2).

In the new version of our manuscript, we have implemented several changes to address these
comments. We also show novel data that critically substantiate our previous findings and
further support the interpretation of our results. These new data have been listed and shortly
presented in our reply to the general feedback from the Editorial Board (see above). Taken
together, we believe that these changes have significantly clarified and improved our
manuscript.

*2.For example, the authors conclude that IFN γ production by NK cells is not relevant to*
*the enhanced CD8 response but they still study this aspect of NK cell function later on in the*
*manuscript.*

Secretion of cytokines, especially IFN γ , represents a key function of NK cells, in particular
during virus infection (1). For this reason, we believe that our data indicating an altered
cytokine profile in NK cells of LCMV-infected *Trail*^{-/-} mice (Figure 3A-C) and our results from
the transcriptomic analysis (Figure 3—figure supplement 1, revised manuscript) warranted
further investigation on the mechanisms underlying these phenotypes and the role of IFN γ in
our model.

Indeed, previous studies have shown that IFN γ signaling is critical to restrict the spread of
various LCMV strains at the onset of infection, therefore promoting the virus-specific CD8⁺ T

cell response and later pathogen clearance (2). Consequently, *Ifng*^{-/-} mice fail to clear a slowly
invasive strain of LCMV (Armstrong strain) (3), and infection of *Ifngr1*^{-/-} mice with a low dose
(10² pfu) of LCMV WE virus leads to a fatal wasting disease (2). LCMV WE is particular
susceptible the antiviral effect of IFN γ (4). Correspondingly, antibody-mediated blockade of
IFN γ of LCMV WE-infected mice has been shown to impair virus clearance and the specific
CD8⁺ T cell response (5).

Nevertheless, it is correct that our data indicate that the higher levels of IFN γ that are produced
by NK cells in LCMV-infected *Trail*^{-/-} mice do not account for the increased CD8⁺ T cell
response in these animals (Figure 3E & F, revised manuscript). This conclusion is further
supported by our latest findings from an *in vivo* NK cell-mediated killing assay revealing that
the specific CD8⁺ T cell response in LCMV-infected *Trail*^{-/-} mice likely derives from an
impairment of the cytotoxic function of NK cells in these mice – rather than from an increased
systematic level of IFN γ , which is unlikely to affect the proliferation of CD8⁺ T cells in such a
short 6 hour assay (Figure 4B-D, revised manuscript). These aspects are presented in the
discussion of the revised manuscript, together with a reference to the former studies
demonstrating the relevance of IFN γ signaling for LCMV immunity.

The reason for still investigating the *Trail*-dependent cellular mechanisms controlling IFN γ
production was to evaluate whether TRAIL may regulate other pathways in NK cells besides
the signaling downstream of IL-15 receptor. Indeed, our results show that TRAIL signaling not
only modulates the IL-15—GZMB axis, but also the NK.1.1—IFN γ axis, and possibly other
pathways related to inflammation and cytokine production or signaling (Figure 5; Figure 5—
figure supplement 2D and Figure 3—figure supplement 1, revised manuscript). These findings
provide evidence for a multifaceted and broad contribution of TRAIL to the regulation of NK
cell function. Yet, we have slightly modified the abstract to reduce possible confusion on the
role of IFN γ for the immune response in LCMV-infected *Trail*^{-/-} mice.

Please note that comments on the role of IFN γ in our model were also raised by Reviewer #2
(major comment #3 and #5) and Reviewer #3 (major comment #4). In our reply to major
comment #5 of Reviewer #2, we further discuss the function of IFN γ in our model (see below).

*3. In addition, a number of conclusions are reached that are not directly supported by the*
*experiments presented. For example, the authors conclude that GZB production is relevant to*
*the phenotype but do not directly test if NK cell cytotoxicity of CD8 T cells is actually*
*operational in their studies.*

To address this relevant comment, we used an *in vivo* NK cell-mediated killing assay to
demonstrate that *Trail*^{-/-} NK cells exhibit reduced cytotoxicity against LCMV-specific CD8⁺ T
cells, when compared to WT NK cells (Figure 4B & C, revised manuscript). Since activated
LCMV-specific CD8⁺ T cells do not express DR5, we conclude that this phenotype is likely
caused by an impaired activation of the IL-15—GZMB pathway in activated *Trail*^{-/-} NK cells,
and not dependent on TRAIL-induced apoptosis (Figure 4D & Figure 5, revised manuscript).
This is in agreement with our data showing that activated *Trail*^{-/-} NK cells have a reduced
ability to eliminate TRAIL-resistant YAC-1 cells *in vitro* (Figure 4A, revised manuscript), as
well as previous studies reporting a central role of the perforin-granzyme pathway for the NK
cell-mediated elimination of LCMV-activated WT (6), transgenic CD8⁺ (P14) T cells (7) and
*Ifnar1*^{-/-} P14 or *Ifnar1*^{-/-} transgenic CD4⁺ (Smarta) T cells (8, 9).

A similar comment was raised by Reviewer #2 (major comment #4) and Reviewer #3 (major
comment #3).

*4. They show many experiments suggesting that TRAIL affects IL15 signaling in NK cells, but*
*other data suggest that the effects in vivo are due to NK cell extrinsic effects.*

We have modified the manuscript text and added novel data to clarify the mechanisms by which
TRAIL signaling regulates NK cell function. In particular we implemented several changes
and improvements in the new version of the manuscript. Taken together, our data show the
following:

- 1. LCMV infection leads to upregulation of TRAIL and TRAIL receptor (DR5) on splenic
mouse NK cells (Figure 2C). These data were imprecisely described in the previous
version of the manuscript (please, also refer to our reply to minor comment #2 of
Reviewer #1 and to minor comment #5 of Reviewer #2).
- 2. *Trail*-deficiency in whole splenocytes or MACS-purified NK cells leads to a reduction
of the signaling induced downstream of IL-15 receptor (Figure 5; Figure 5—figure
supplement 1A). This is also discussed in our reply to minor comment #3 of Reviewer
#2.

- 3. TRAIL blockade of activated NK cells represses the signaling downstream of IL-15
 receptor. This is true for murine and human primary NK cells, as well as for the human
 NK cell line NK-92 (Figure 5F-I; Figure 5—figure supplement 1D & E; Figure 6;
 Figure 7—figure supplement 1D & E).
- 4. Inversely, addition of plate-coated recombinant TRAIL promotes S6 phosphorylation
 downstream of IL-15 receptor in NK cells. However, addition of soluble TRAIL does
 not recapitulate this ancillary effect on IL-15 signaling (Figure 7E, revised manuscript).
- 5. Provision of membrane-bound TRAIL by adding WT splenocytes to IL-15-stimulated
 *Trail*^{-/-} splenocytes restores the IL-15—GZMB axis in *Trail*^{-/-} NK cells, in a dose-
 dependent manner. In contrast, addition of *Trail*^{-/-} splenocytes to reduce TRAIL
 availability results in a decreased activation of IL-15 signaling in WT NK cells (Figure
 7A & B). A similar effect can be also observed in *in vivo* conditions (Figure 7C & D).

Collectively, these data indicate that cell-extrinsic, membrane-bound TRAIL engages TRAIL
 receptor on NK cells to modulate the signaling downstream of IL-15 receptor. Impaired
 activation of the IL-15 pathway in activated *Trail*^{-/-} NK cells leads to lower granzyme B
 expression and diminished cytotoxicity, thereby likely resulting in reduced elimination of
 virus-specific CD8⁺ T cells in LCMV-infected *Trail*^{-/-} mice. Figure 7—figure supplement 2
 shows a schematic model of this previously unrecognized regulatory function of TRAIL
 signaling in NK cells.

*5. That TRAIL appears to affect IL15 signaling in NK cells is quite interesting and apparently*
 *novel but there are no experiments to address how this occurs.*

We believe that we provide several lines of evidence indicating that IL-15 signaling and non-
 canonical TRAIL signaling converge to activate the PI3K—AKT—mTOR pathway and thus
 synergize to regulate NK cell function. We briefly review below previous studies describing
 non-canonical TRAIL signaling in different types of (transformed) cells and the current
 knowledge on IL-15 signaling in NK cells. We then present our novel findings in light of these
 studies, which are presented in the revised manuscript.

Previous studies on non-canonical TRAIL signaling and on IL-15 signaling in NK cells

Non-canonical signaling downstream of TRAIL receptor activates PI3K in various TRAIL-
 resistant cancer cell lines, leading to AKT activation and thus promoting cell survival (instead
 of apoptosis) or migration (reviewed in (10, 11)). This has been reported for Jurkat leukemia T

cells (12), human breast cancer and ovarian cancer cells (13), non-small cell lung cancer cells
(14) and prostate adenocarcinoma cells (15). Human umbilical vein endothelial cells
(HUVECs) also react to TRAIL by activating PI3K and AKT, which prevents apoptosis
induced by trophic withdrawal (16). While this non-canonical TRAIL—PI3K—AKT signaling
is established, its precise underlying molecular mechanisms are still unclear (17).

IL-15 promotes NK cells either via activation of the JAK/STAT pathway for the transcription
of pro-survival genes during homeostatic conditions (18, 19), or via induction of the PI3K—
AKT—mTOR axis upon activation, to support the development of effector functions and
GZMB protein expression (20-22).

Data in the revised manuscript indicating that TRAIL contributes to IL-15 signaling

Using primary NK cells from genetically-modified mice, we demonstrate that *Trail*-deficiency
leads to a reduction of the PI3K—AKT—mTOR signaling induced downstream of IL-15
receptor. TRAIL blockade, employing TRAIL-R2-Fc chimeric protein, results in a similar
effect (Figure 5F-I; Figure 5—figure supplement 1A, D &E; Figure 6; Figure 7—figure
supplement 1D & E, revised manuscript). Inversely, plate-coated recombinant TRAIL
promotes S6 phosphorylation downstream of IL-15 receptor in NK cells (Figure 7E, revised
manuscript).

These data indicating a link between TRAIL signaling and the IL-15—PI3K—AKT—mTOR
axis were further corroborated by a transcriptomic analysis which revealed that several
pathways related to PI3K/AKT signaling and IL-2 family signaling (to which IL-15 belongs)
are differently affected in WT versus *Trail*^{-/-} NK cells upon LCMV infection (Table 1 and Table
2, revised manuscript).

In addition, we also found that pharmacological inhibition of PI3K using the classic PI3K
inhibitors wortmannin/LY294002 abolishes the promoting effect of TRAIL signaling on IL-
15-induced GZMB expression in WT NK cells. In other terms, PI3K inhibition leads to similar
GZMB expression in IL-15-stimulated *Trail*^{-/-} versus WT NK cells (**PBP-Figure 1A** and
Figure 5—figure supplement 1C, revised manuscript). Accordingly, PI3K inhibition with
wortmannin or LY294002, given individually or in combination, decreases pAKT levels in IL-
15-stimulated WT NK cells to the levels measured in *Trail*^{-/-} NK cells after the same treatment
(data not shown). Taken together, these results imply that PI3K is an intersection point in the
signaling downstream of TRAIL receptor and IL-15 receptor (**PBP-Figure 1B-D**). Of note, our

results and their interpretation concur with the findings from another study indicating that
TRAIL can activate a PI3K—AKT—mTOR axis in TRAIL-resistant cancer cell lines (13).

Collectively, these data show that TRAIL signaling promotes the IL-15—PI3K—AKT—
mTOR—GZMB axis in NK cells. In the discussion of the revised manuscript, we compare
these previous studies describing non-canonical TRAIL—PI3K—AKT signaling in TRAIL-
resistant cancer cell lines with our findings in primary NK cells that indicate that the pathways
downstream of IL-15 receptor and TRAIL receptor appear to converge by activating the
PI3K—AKT—mTOR—GZMB pathway.

**PBP-Figure 1. TRAIL signaling activates PI3K to promote in NK cells the pathway downstream**
 **of IL-15 receptor.** (A) Wortmannin and LY294002 were used at the concentrations recommended by
 the manufacturer (1 μ M and 50 μ M, respectively) and added to splenocyte cultures 1 hour before addition
 of IL-15. GZMB expression was measured in NK cells by flow cytometry after 20 more hours. Data
 shown are mean \pm SEM of $n=3$ mice per group. Statistical analysis was performed using one-way
 ANOVA with Dunn's post-test. * $p<0.05$; ** $p<0.01$. This panel is identical to the panel displayed in
 Figure 5—figure supplement 1 of the revised manuscript. Model showing the interaction between non-
 canonical TRAIL signaling and IL-15 signaling in NK cells. (B) Non-canonical TRAIL signaling
 activates PI3K in defined cell types (10, 11), and IL-15 signals in NK cells through a PI3K—AKT—
 mTOR pathway to promote the expression of GZMB protein (20–22). If these two pathways are
 engaged in activated NK cells, non-canonical TRAIL signaling activates PI3K to promote the IL-15—
 PI3K—AKT—mTOR—GZMB pathway. (C) Disruption of TRAIL signaling via genetic deletion or
 the use of a blocking compound results in reduced signaling downstream of IL-15 receptor in activated
 NK cells. (D) Wortmannin and LY294002 inhibit PI3K, thereby blocking non-canonical TRAIL
 signaling and the IL-15—mTOR pathway in activated NK cells.

Interestingly, and in agreement with a previous report (23), IL-15 stimulation promotes the
expression of TRAIL and TRAIL receptor (DR5) on murine NK cells (**PBP-Figure 2**), which
suggests a feed-forward loop between IL-15 and TRAIL signaling.

**PBP-Figure 2. IL-15 stimulation promotes the expression of TRAIL and TRAIL receptor on**
**murine NK cells.** WT splenocytes were stimulated with IL-15 and expression of (A) TRAIL or (B)
TRAIL receptor (DR5) were assessed by flow cytometry at the indicated time points. Representative
histograms are shown (n=2 per time point). Unst., unstimulated.

**Reviewer#1 / Minor comments:**

*1. Fig 1C. Not clear why two asterisks at day 8 and one at day 10 when there are minimal*
 *effects on titers.*

We carefully cross-checked and recalculated the statistics showed in Figure 1C, and they are
 indeed correct. In **PBP-Figure 3**, we present a selection of the results of Figure 1C, which are
 shown in greater details for day 8, 10 and 12 post LCMV infection – i.e. the time points where
 differences in virus clearance can be observed between WT and *Trail*^{-/-} mice. The indicated p
 values for day 8 and day 10 are correct (p=0.0028 and p=0.0334, respectively).

In addition, for each of the compared group n=3 for day 7 (p=0.1) while n=9 for day 8
 (p=0.0028), which may explain why there are only statistically significant differences for the
 titers in WT versus *Trail*^{-/-} spleens from day 8 on.

**PBP-Figure 3. Improved virus control in *Trail*-deficient mice.** WT and *Trail*^{-/-} were infected with
 LCMV and virus titers were determined in spleen at the indicated time points post infection. The limit
 of detection (LOD) is represented by a dotted horizontal line. Data indicate mean ± SEM pooled data
 from 1-4 independent experiments. Statistical analyses were performed using Mann-Whitney test and
 p values are shown. This figure is a detailed representation of the Figure 1C of the manuscript, for days
 8-12.

The group size for each of the days indicated in Figure 1C is as follows, for the spleen: day 1,
 n=9; day 2, n=9; day 3, n=3; day 5, n=3; day 7, n=3; day 8, n=9; day 10, n=11; day 12, n=6.

The virus titer in Figure 1C is plotted on a log scale – a common representation of this type of
 data – which may lead to some confusion. We have therefore added in the new version of the

manuscript more detailed information on group size for the virus titer (Materials and Methods,
section on “virus and virus titration”).

Please note that a similar comment was raised by Reviewer #2 (minor comment #4).

*2. Fig 2C. Not clear what cells are being studied.*

In Figure 2C, we measured TRAIL and TRAIL-R (DR5) surface protein expression on splenic
NK cells. This information has now been added to the figure legend.

*3. Line 126. Premature to conclude that TRAIL is affecting NK cell activity.*

The following sentence is referred to:

“Indeed, LCMV triggered upregulation of TRAIL and TRAIL receptor (TRAIL-R or
DR5, which is encoded by *Tnfrsf10b*) on NK cells (Figure 2C), thus indicating a potential
effect of TRAIL/TRAIL-R signaling on NK cell activity during LCMV infection.”

We respectfully disagree with this comment from Reviewer #1; in this sentence we do not
conclude that TRAIL/TRAIL-R signaling affects NK cell activity. We rather indicate that the
data displayed in Figure 2A-C (revised manuscript) suggest a **potential** effect of the TRAIL
pathway on NK cell activity (which we show later in Figures 3-7).

*4. Fig 2D. Should be replotted to show anti-NK1.1 next to untreated for WT, and for TRAIL-*
*def.*

Several studies have previously addressed the effect of NK cell depletion on the T cell response
during LCMV infection (6, 7, 24). These studies discriminated between presence and absence
of NK cells.

However, the current study aims at evaluating the effect of TRAIL during LCMV infection by
directly comparing WT and *Trail*^{-/-} mice side-by-side. In this context, we find that NK cell
depletion abolishes the *Trail*-dependent limiting effect on the T cell response, thus resulting in
similar frequencies of (IFN γ -producing) specific CD8⁺ T cells in WT and *Trail*^{-/-} groups after
NK cell depletion (Figure 2D and Figure 4C, revised manuscript). Therefore we believe that
the current representation of the data, which juxtaposes the WT and *Trail*^{-/-} groups to allow
direct group comparisons in conditions where NK cells are present or depleted, is more
appropriate for the purpose our study.

**Comments of Reviewer #2:**297 **Reviewer #2 / Major comments:**

*1. This manuscript describes a non-canonical role for TRAIL in determining NK-cell function*
 *during viral infection. They conclude that cell-extrinsic TRAIL engages TRAIL-R on NK cells,*
 *supports IL-15 signaling and promotion of granzyme B while reducing IFN-gamma.*
 *Consequently, NK cells suppress CD8 T cells and alter viral pathogenesis. The experiments*
 *appear sound and represent important findings that will be intriguing to the scientific*
 *community. However, several issues reduce confidence in the interpretation of the data and*
 *potentially undermine the mechanistic conclusions.*

We thank Reviewer #2 for the careful reading of our manuscript and for rightly identifying
 weaknesses in the interpretation of our data. As presented in our reply to the general feedback
 from the Editorial Board (see above), we have revised our manuscript and added novel data.
 We believe that these changes have corroborated our initial conclusions.

*2. What is the evidence that the phenomenon is independent of pro-apoptotic functions of*
 *TRAIL? Apoptosis is only examined in Figure 5.S2, where sTRAIL does not trigger apoptosis*
 *in NK cells. Yet, the authors don't delineate which specific cell express TRAIL or TRAIL-R,*
 *thereby precluding determination of apoptosis in the relevant lineage. The role of membrane-*
 *bound TRAIL rather than soluble TRAIL is never formally tested. The non-apoptotic conclusion*
 *must be supported by data on the ability/inability of TRAIL or TRAIL-R engagement by soluble*
 *or membranous ligands to induce NK cell apoptosis. Moreover, the potential for TRAIL-*
 *mediated apoptosis of accessory cells that subsequently influences NK cell function should be*
 *discussed.*

To address this relevant comment, we have assessed the expression of TRAIL and its receptor
 DR5 on different immune cell populations in murine spleen. We found that TRAIL and DR5
 proteins become upregulated on NK cells 24 hours after LCMV infection (Figure 2C, revised
 manuscript). TRAIL was also expressed at the same time point post infection on CD3⁺, CD4⁺
 or CD8⁺ T cells and on monocytes (defined as CD11b⁺CD11c⁻Ly6C⁺Ly6G⁻ cells). However,
 TRAIL was not detectable on dendritic cells (defined as CD11c⁺MHCII⁺Ly6C⁻Ly6G⁻,
 CD11c⁺MHCII⁺Ly6C⁻Ly6G⁻CD8⁺ or CD11c⁺MHCII⁺Ly6C⁻Ly6G⁻CD8⁻ cells) and on
 neutrophils (defined as CD11b⁺CD11c⁻Ly6G⁺) (data not shown). Among these different cell

populations, DR5 was only expressed on NK cells and monocytes (Figure 2C and data not
shown). Importantly, at 6 days post infection, DR5 is not expressed on activated LCMV-
specific CD8⁺ T cells (Figure 4D, revised manuscript). This strongly implies that these cells
cannot engage TRAIL signaling. We therefore conclude that the NK cell-mediated elimination
of virus-specific CD8⁺ T cells in our model is likely independent of TRAIL pro-apoptotic
function.

Therefore, while many immune cells express TRAIL during early LCMV infection, few of
these cells are able to engage TRAIL signaling downstream of DR5. These findings, together
with the fact that experiments using MACS-purified primary NK cells or NK cells lines
recapitulate the phenotypes observed *in vivo*, do no hint to a possible effect on NK cell function
of accessory immune cells in which apoptosis was induced via activation of TRAIL signaling.
Yet it is also correct that we cannot formally prove this, and we therefore discuss this aspect in
the revised manuscript. Moreover, our data on TRAIL and DR5 expression on immune
populations before and after LCMV infection are now either referred to or shown in the revised
manuscript.

To further address this comment from Reviewer #2, we also performed *in vitro* experiments to
evaluate which form of TRAIL modulates the function of NK cells. Noteworthy, our data
indicate that the non-canonical, regulatory contribution of TRAIL signaling for NK cell
function is operative only in activated NK cells, as basal levels of phosphorylated AKT and S6
are comparable in naïve *Trail*^{-/-} versus WT NK cells (Figure 5—figure supplement 1B, revised
manuscript). Indeed, DR5 is not expressed on naïve splenic murine NK cells, yet it becomes
upregulated upon LCMV infection both on WT and *Trail*^{-/-} NK cells (Figure 2C, revised
manuscript). To enable NK cells to engage TRAIL signaling *in vitro*, we stimulated these cells
with IFN α , which has been reported to promote DR5 expression (25). We found that prolonged
*in vitro* treatment with IFN α was necessary for NK cells to express DR5 (**PBP-Figure 4A and**
**B**). We then incubated IFN α -treated *Trail*^{-/-} NK cells with different concentrations of plate-
coated versus soluble TRAIL to compensate for the lack of TRAIL in these cultures. Addition
of recombinant TRAIL, whether plate-bound or soluble, did not affect the ability of *Trail*^{-/-} NK
cells to respond to IL-15 stimulation after short-term IFN α treatment (**PBP-Figure 4C**) – which
was expected due to the absence of DR5 expression on NK cells in these conditions (**PBP-**
**Figure 4A**). However, while prolonged IFN α treatment upregulated DR5 expression on NK
cells, it also rendered these cells hypo-sensitive to IL-15 (**PBP-Figure 4B and D**), which

prevented us to discriminate a possible differential role of membrane-bound/plate-coated
 versus soluble TRAIL on IL-15 signaling in primary NK cells.

**PBP-Figure 4. Prolonged stimulation with type I interferon promotes DR5 expression on NK cells,**
 **yet it renders them hyporesponsive to IL-15 stimulation.** Whole splenocytes were pooled from 3
 *Trail*^{-/-} mice and treated with IFN α (150ng/ml) for (A) 24 hours and (B) 4 days, and DR5 expression
 was measured on NK cells using flow cytometry. Alternatively, *Trail*^{-/-} splenocytes were first treated
 for (C) 24 hours or (D) 4 days with IFN α and then stimulated with different concentrations of
 recombinant plate-bound or soluble TRAIL (for a total of 1.5 hours) and IL-15 (for 1 hour) (n=1 per
 condition). Flow cytometry was then applied to measure S6 phosphorylation in NK cells. Data indicate
 show (A, C) one representative of two independent experiments. (B, D) One experiment was performed.
 Similar data were generated using splenocytes from WT mice.

To overcome these technical limitations, we next used the NK-92 human cell line that shows
 constitutive expression of human TRAIL and its receptors DR4 and DR5, and which is
 therefore endowed with the capacity to engage TRAIL signaling. Of note, NK-92 cells were
 found to behave similarly to primary murine and human NK cells, since TRAIL blockade also
 repressed the signaling downstream of IL-15 receptor in this cell line (Figure 7—figure
 supplement 1D, E). Remarkably, addition of plate-coated – but not soluble – recombinant
 TRAIL promoted, in a dose-dependent manner, the signaling downstream of IL-15 receptor in
 NK-92 cells (Figure 7E, revised manuscript).

Taken together, these findings support the notion that cells expressing membrane-bound
 TRAIL trigger TRAIL signaling in TRAIL-R-positive NK cells, which in turn promotes IL-15
 signaling, granzyme B expression and NK cell cytotoxicity. Therefore, our data imply that
 soluble TRAIL does not appear to regulate NK cell function.

3. *The current presentation creates unnecessary confusion about the directionality and cell-*
*intrinsic nature of events. None of the data support the necessity or sufficiency of TRAIL-R*
*engagement on NK cells driving these effects, while Fig5.S2 casts some doubt on this*
*hypothesis. The one experiment clearly performed with isolated NK cells (Fig5.S3) suggests*
*that Trail-/- NK cells are defective in response to NK1.1 stimulation, which stands in contrast*
*to the conclusion that this is mediated via TRAIL-R engagement. Analysis of NK-cell expression*
*of IFN-gamma and granzyme B in Figure 7C/D, or in vitro experiments comparing effect of*
*sTRAIL/mTRAIL/TRAIL-R stimulation on these functions of NK cells, would boost confidence*
*in the conclusions.*

Directionality and cell-intrinsic nature of events

The issue on the cell-autonomous versus cell-extrinsic function of TRAIL has been addressed
in our reply to major comment #4 of Reviewer #1.

In vitro experiments comparing effect of soluble TRAIL/membrane-bound TRAIL/TRAIL-R 397 stimulation

Our novel data comparing the differential contribution to NK cell regulation of membrane-
bound versus soluble TRAIL were presented above in our reply to major comment #2 of
Reviewer #2 (Figure 7E, revised manuscript).

Analysis of NK cell expression of IFN γ

Because our data do not provide evidence for a contribution of NK cell-derived IFN γ to the
CD8⁺ T cell phenotype detected in LCMV-infected *Trail*^{-/-} mice, and to follow the opinion of
Reviewer #1 (major comment #2), we did not explore the TRAIL-dependent mechanisms
regulating IFN γ expression in NK cells beyond the investigations carried out and presented in
the first version of our manuscript.

Analysis of NK cell expression of granzyme B

For the analysis of *Trail*-dependent S6 phosphorylation, splenic NK cells were analyzed 24
409 hours after LCMV infection (Figure 5E, revised manuscript). For the experiment presented in
Figure 7D (revised manuscript) – which investigates the contribution of different TRAIL
availabilities in the host environment to the *in vivo* activation of transferred NK cells – flow
cytometry analysis of S6 phosphorylation was performed 48 hours after adoptive transfer and
24 hours after LCMV infection (see detailed experimental setup in Figure 7C, revised
manuscript).

In comparison, GZMB expression in NK cells was (logically) assessed later than S6
 phosphorylation, i.e. 5 days post infection (Figure 4F, revised manuscript). To evaluate GZMB
 expression in NK cells in a setting comparable to the one presented in Figure 7C/D (revised
 manuscript), we first adoptively transferred congenic splenocytes into recipient mice and
 infected these animals with LCMV 24 hours after cell transfer. NK cells were then analyzed
 by flow cytometry 24 hours or 5 days post infection (i.e. 2 or 6 days after cell transfer,
 respectively) (**PBP-Figure 5A**). Compared to the test at the first time point (day 1 post
 infection), analysis at the later time point (day 5 post infection) did not allow for donor NK
 cells to be recovered in amounts sufficient for reliable GZMB measurement (**PBP-Figure 5B
 and C**). Therefore, this technical limitation prevents us from directly addressing this specific
 comment of Reviewer #2.

Note that a similar remark about GZMB and IFN γ expression after adoptive transfer (in the
 experimental setup showed in Figure 7C and D) was raised by Reviewer #3 (major comment
 #4).

**PBP-Figure 5. Analysis of host and donor NK cells in LCMV-infected mice.** (A) Experimental setup
 of WT (Ly5.1) donor splenocyte transfer (10^7 cells) into WT (Ly5.2) recipient mice followed by LCMV
 infection. Ratios of donor and host splenic NK cells were quantified by flow cytometry on day 1 (B) or
 on day 5 (C) post LCMV infection. Data are shown for two individual mice per time point and are
 representative of one of 2 independent experiments.

4. The conclusion that lower *GzmB* expression results in reduced suppression of CD8 T cells
is tenuous. Do the authors have any data that support a role for *GzmB* in suppression? The
authors should test that TRAIL-deficient, *gzmB*-low NK cells exhibit reduced capacity to
suppress T cells (in vitro perhaps). Ideally, the authors would also show that suppressive
function can be rescued by restoring *gzmB* expression in *Trail*^{-/-} NK cells, or mimic effect by
knocking-down *GzmB* in wild-type NK. Given the difficulty of the latter experiments, a
thorough discussion of the limits of the data should be provided at a minimum.

To address this pertinent comment for additional evidence for the reduced ability of *Trail*^{-/-} NK
cells to eliminate activated CD8⁺ T cells (in a GZMB-dependent manner), we have followed
the following approaches, which all turned out to be technically challenging due to the reasons
indicated below:

**Setup #1.** An *ex vivo* killing assay was first set up using MACS-purified NK cells isolated from
the spleen of LCMV-infected *Trail*^{-/-}, WT and *Prf*^{-/-} mice, which were co-incubated with
MACS-purified P14 cells isolated from NK cell-depleted and LCMV-infected WT recipient
mice. However, this approach failed to disclose differences in the cytotoxic capacity of the NK
cells of these different strains (data not shown), thereby confirming the data from previous
reports indicating that P14 cells are resistant to (perforin-mediated) (WT) NK cell-mediated
killing in such *ex vivo* settings (8, 9).

**Setup #2.** An *in vitro* assay was performed in which quantification of T cell proliferation was
used as a surrogate of NK cell-mediated killing, as previously applied (7). For this purpose,
MACS-purified WT CD8⁺ T cells were activated *in vitro* with anti-CD3/CD28 antibodies and
co-incubated with WT or *Trail*^{-/-} MACS-purified NK cells from naïve or d5 LCMV-infected
mice. Fluorescence dilution was then measured in CD8⁺ T cells on day 1, 2 and 3. This assay
did not show differences in the ability of *Trail*^{-/-} versus WT NK cells to restrict *in vitro* CD8⁺
T cell proliferation. There was also no differences between the conditions using naïve versus
LCMV-primed NK cells (data not shown).

**Setup #3.** Another *in vitro* cytotoxicity assay was also set up in which MACS-purified *Trail*^{-/-}
and WT NK cells were first activated *in vitro* with IL-2 for 4 days and then co-incubated with
WT CD8⁺ T cells that had been previously activated for 24h or 48h with concanavalin A
(ConA). Comparison with the control (i.e. experimental conditions using *Prf*^{-/-} NK cells) did
not show evidence for a cytotoxic activity of NK cells in this setting, as assessed by annexin V
expression on target T cells (data not shown). This is in agreement with the conclusions of the

data from the *ex vivo* killing assay presented above (set up #1) indicating that activated WT
CD8⁺ T are poorly susceptible to (WT) NK cell-mediated cytotoxicity in such *in vitro* settings.

**Setup #4.** A similar *in vitro* cytotoxicity assay as presented in setup #3 was also applied, yet
using anti-CD3/CD28 antibody-activated *Ifnar1*^{-/-} CD8⁺ T cells as NK cell targets, since
activated T cells incapable of engaging type I IFN are particularly susceptible to
perforin/granzyme-dependent NK cell-mediated lysis (8, 9). However, in these settings we
again did not detect differences between the conditions using IL-2-activated *Trail*^{-/-}, WT or *Prf*
474 ^{-/-} NK cells (data not shown).

**Setup #5.** As a variation of the assay presented in setup #4, we next tried to use as NK cell
targets MACS-purified *Ifnar1*^{-/-} P14 cells isolated from NK cell-depleted and LCMV-infected
WT recipient mice. However, the limited quantity of recovered target cells in these settings –
which was previously also observed by other researchers (Annette Oxenius, personal
communication) – prevented us to perform an *in vitro* cytotoxicity assay with these LCMV-
primed *Ifnar1*^{-/-} P14 cells.

**Setup #6.** An *in vivo* cytotoxicity assay was also carried out, in which P14 cells were isolated
from NK cell-depleted and LCMV-infected WT recipient mice and re-transferred into NK cell-
replete or NK cell-depleted LCMV-infected *Trail*^{-/-} versus WT mice. Frequencies of P14 cells
were then measured in the spleen of recipient mice 6h after adoptive transfer. In these
conditions, there were again no differences in the frequency of recovered target P14 cells
between the groups, in line with a previous study showing that P14 cells are resistant to NK
cell-mediated killing in such *in vivo* settings (8).

**Setup #7.** At last, we performed an *in vivo* cytotoxicity assay comparable to the one presented
in setup #6, yet using MACS-purified *Ifnar1*^{-/-} P14 cells as NK cell targets (since, as mentioned
above, these cell are particularly sensitive to perforin/granzyme-dependent NK cell-mediated
elimination (8, 9)). In these conditions, we found a higher recovery of *Ifnar1*^{-/-} P14 cells from
infected *Trail*^{-/-} versus WT mice. Furthermore, NK cell depletion in these settings led to an
even higher target cell recovery and abolished the strain-specific difference (Figure 4B & C,
revised manuscript). Collectively, these results are in agreement with the NK cell-dependent
differences in endogenous CD8⁺ T cell priming observed upon LCMV infection of *Trail*^{-/-}
versus WT mice (Figure 1A and Figure 2D, revised manuscript).

Although our data do not directly demonstrate a role of GZMB for the NK cell-mediated
elimination of target CD8⁺ T cells during LCMV infection – addressing this aspect would be

technically challenging, also considering the difficulties we encountered and that are presented
 above – they nevertheless strongly support this notion. Indeed, perforin and granzyme act
 synergistically (26), and our study indicates that virus-primed LCMV-specific CD8⁺ T cells do
 not express DR5 and therefore cannot engage the (apoptotic) signaling downstream of TRAIL
 receptor (Figure 4D, revised manuscript). In addition, activated *Trail*^{-/-} NK cells show reduced
 GZMB expression and have an impaired ability to kill TRAIL-resistant YAC-1 cells *in vitro*
 (Figure 4F and Figure 4A, respectively; revised manuscript). Yet we implemented this remark
 of Reviewer #2 by making several changes in the manuscript, where we now indicate that
 reduced GZMB expression in activated *Trail*^{-/-} NK cells is *associated with* or *linked to* (rather
 than a direct cause of) reduced (TRAIL-independent) cytotoxic function.

*5. The role of enhanced IFN-gamma is incompletely addressed. Figure 2B (24 hours), Fig 2-*
 *S2 (36 h), and Fig 3 (24-48h) show early IFN-gamma expression, but enhancement of IFN-*
 *gamma at later time points is not examined. The possibility that continued elevations in IFN-*
 *gamma levels can contribute undermines the conclusions of Fig 3F that is purported to show*
 *effects on CD8 T cells are independent of early IFN-g. A better experiment would be*
 *administration of anti-IFN-gamma blocking antibodies during initial days of infection to*
 *reverse (or not) enhanced CD8 T cell responses in Trail^{-/-}.*

Published work on the role of IFN γ for the immune response to LCMV

We do agree with this comment from Reviewer #2 that IFN γ may contribute to the CD8⁺ T cell
 phenotype observed in LCMV-infected *Trail*^{-/-} mice. In our reply to the major comment #2 of
 Reviewer #1, we mention several reports that have demonstrated the central role of IFN γ
 signaling to LCMV control and the induction of the virus-specific CD8⁺ T cell response (2-5).
 We also discuss this aspect in the revised manuscript.

Role of IFN γ in infected *Trail*^{-/-} mice at later time points

As shown in Figure 3A, the IFN γ production in NK cells is the highest 24 hours post LCMV
 infection and it then drops considerably one day later. IFN γ becomes undetectable in NK cells
 36 hours post LCMV infection (data not shown). These data are in agreement with a previous
 study reporting a maximum of IFN γ production in peritoneal NK cells 30 hours after LCMV
 infection, which then immediately decreases (27).

We have not assessed the NK cell-specific IFN γ production at later time points during the
infection. It likely that serum IFN γ levels will raise at later time points in LCMV-infected
(*Trail*^{-/-}) mice, to reach a maximum before the peak of the T cell response (which occurs
between day 6 and day 8 post infection). As a matter of fact, a former study has shown that
LCMV WE infection induces two waves of IFN γ production in the serum of WT mice; a first,
smaller peak on day 2 and a second, larger peak on day 5 of the infection (28). Another report
mentioned the IFN γ serum level of LCMV WE-infected mice to be the highest between day 4
and day 8 post infection. While in this study NK cell-depletion did not affect levels of IFN γ in
serum and spleen at day 4, CD8⁺ T cell-deficient mice showed strong reduction in systemic
and splenic IFN γ production (29).

Taken together, LCMV induces IFN γ the first days after infection mainly in NK cells and later
in CD8⁺ T cells. Since higher frequencies of IFN γ -positive CD8⁺ T cells are found in infected
*Trail*^{-/-} mice, it is very likely that these animals will also show higher systemic levels of IFN γ
at the peak of the T cell response.

Suggested IFN γ blockade studies

While the findings from previous studies justify our endeavor to address the role of the NK
cell-produced IFN γ to the improved CD8⁺ T cell response observed in LCMV-infected *Trail*^{-/-}
mice (2, 5), we believe that the IFN γ blockade studies suggested by Reviewer #2 would be less
informative about the mechanisms underlying this phenotype than the experiment presented in
Figure 3E and F (revised manuscript), for the following two main reasons:

1. Role of IFN γ produced specifically by NK cells

Our data in Figure 3 (revised manuscript) indicate that absence of TRAIL signaling during
LCMV infection increases the ability of NK cells to produce IFN γ . Figure 2D shows that NK
cells underlie the increased virus-specific CD8⁺ T cell response in LCMV infected *Trail*^{-/-} mice.
Building on these findings, the purpose of the experiment depicted in Figure 3E and Figure 3F
is to address the role of the IFN γ that is specifically secreted by NK cells to the increased
LCMV-specific CD8⁺ T cell response in infected *Trail*^{-/-} mice. IFN γ blockade studies, as
proposed by Reviewer #2, would likely impact on the T cell response to LCMV, as previously
found by others (5). However, they would not allow to discriminate the effects, in our model,
of NK cell-derived IFN γ from IFN γ derived from other cells, including CD8⁺ T cells. Indeed,

it is unclear how long the effect of an IFN γ blocking antibody would last after administration
at an early stage of LCMV infection.

2. Role of the NK cell cytotoxic function for the higher T cell response in infected
*Trail*^{-/-} mice

Our new data from the NK cell-mediated *in vivo* killing assay provide evidence for the
relevance of the perforin/granzyme-dependent cytotoxic function of NK cells to the T cell
phenotype in infected *Trail*^{-/-} mice (Figure 4B & C, revised manuscript). It is unlikely that the
short duration of this assay (6 hours) would allow for the higher systemic levels of IFN γ to
impact on the improved recovery of target *Ifnar1*^{-/-} P14 cells from infected *Trail*^{-/-} mice.
Nevertheless, the increased IFN γ in infected *Trail*^{-/-} mice may affect target *Ifnar1*^{-/-} P14 cells
either directly or indirectly:

A direct effect of IFN γ on target cells is unlikely. Indeed, lack of IFN γ receptor on CD8⁺ T
cells was reported not to negatively affect the expansion of polyclonal LCMV-specific CD8⁺
T cells *in vivo* (30). This indicates that a direct effect of IFN γ is not required for the priming
of CD8⁺ T cells during LCMV.

An indirect effect of IFN γ on target CD8⁺ T cells is also unlikely. It may for instance affect the
quantity of virus-derived antigens presented on MHC I molecules or the quality of the antigen
presentation. However, there was no difference in the virus load between WT and *Trail*^{-/-} mice
at the time point of the assay (day 4; Figure 1C and Figure 4B, revised manuscript). In addition,
we did not find differences in cell frequencies, total numbers or expression of activation
markers for dendritic cells in the early phase of the infection (Figure 3—figure supplement 3).

Therefore, our data support the notion that the improved specific CD8⁺ T cell response in
LCMV-infected *Trail*^{-/-} mice derives from a reduced perforin/granzyme-dependent cytotoxic
function – and not from an increased IFN γ secretion – of NK cells. This aspect is discussed in
the revised manuscript.

**Reviewer #2 / Minor comments:**

*1. Were proper control antibodies (for anti-NK1.1 or asialoGM1) administered to the non-*
 *depleted group? If not, then additional experiments should be performed with proper controls.*
 *How were the doses of anti-NK1.1 and anti-asialoGM1 selected? Was NK cell depletion*
 *verified? Please change text of Methods, Results, and/or Figure legends to denote answers to*
 *these inquiries.*

For studies involving NK cell depletion, we injected saline (PBS) as a negative control. The
 dose of the anti-NK1.1 antibody (clone PK136 (31)) was selected based on previous
 publications (32-34). For the anti-asialo GM1 antibody, we followed the manufacturer's
 recommendations for the dose to be applied. Importantly, the efficacy of NK cell depletion was
 verified and found to be thorough (**PBP-Figure 6**).

We have accordingly further detailed the Materials and Methods, section on "NK cell
 depletions".

**PBP-Figure 6. Assessment of the efficacy of the used antibody-mediated NK cell depletion**
 **protocol.** Mice were injected with NK cell-depleting antibody or saline (PBS) and LCMV. After 24
 600 hours, single-cell suspensions were prepared from spleens and analyzed by flow cytometry. Each flow
 cytometry plot represents a distinct animal. Treatment conditions and NK cell frequencies in the spleen
 are indicated in the graph. **(A)** NK cell depletion was performed using 200 μ g of anti-NK1.1 antibody
 injected i.p. on day -3 and day -1 (aNK1.1; clone PK136; lower panels). **(B)** NK cell depletion was
 performed using 20 μ l of anti-asialo GM1 antibody injected i.p. on day -1 (aGM1; lower panels).

2. The authors assert that the partial phenotype of *Trail*-deficiency, relative to NK-depletion,
reflects overlapping roles of NK cells and *Trail*. This could just as likely represent a separate
effect of NK cell depletion (like enhanced CD4) that masks or overrides the *Trail* effect.
Detailed discussion of potential interpretations is warranted.

It is indeed possible that NK cell depletion may induce a shift in other immune cell populations,
which may eventually impact the effect of TRAIL on the CD8⁺ T cell response that can be
otherwise observed in NK cell-replete mice. However, we think that this is unlikely to be the
case for the following reasons:

1. In the experimental setup applied, TRAIL appears not to affect the priming of LCMV-
specific CD4⁺ T helper cells (compared to antigen-specific CD8⁺ T cells), and NK cell
depletion similarly increased the expansion and cytokine-secreting activity of these CD4⁺ T
cells in LCMV-infected *Trail*^{-/-} versus WT mice (Figure 2—figure supplement 2).

2. Besides CD4⁺ T cells, dendritic cells (DCs) also represent a critical cell type for the priming
of NK cell-regulated CD8⁺ T cell responses (24, 33). Yet, we did not detect differences in the
phenotype or in the frequencies and numbers of DCs in LCMV-infected *Trail*^{-/-} versus WT
mice (Figure 3—figure supplement 3). Although we have not directly tested it, we believe that
considering these findings it is unlikely that depletion of NK cells may differently alter the
quality or quantity of DC populations in infected *Trail*^{-/-} versus WT animals.

3. We show now that *Trail*^{-/-} NK cells are impaired in their ability to eliminate antigen-specific
CD8⁺ T cells *in vivo* (Figure 4B & C, revised manuscript), and that TRAIL pro-apoptotic
function does not appear to be involved in this NK cell-mediated control of the LCMV-specific
CD8⁺ T cell response (Figure 4D, revised manuscript). Since the *in vivo* NK cell-mediated
killing assay displayed in Figure 4B & C (revised manuscript) is of relatively short duration (6
629 hours), it is unlikely that cells other than NK cells control antigen-specific CD8⁺ T cells in this
setting. In addition, depletion of NK cells in this *in vivo* NK cell-mediated killing assay
abrogated the difference between *Trail*^{-/-} and WT mice in terms of CD8⁺ T cell control (Figure
4C, revised manuscript). These data are in agreement with the results presented in Figure 2D
(revised manuscript) and their interpretation, and they argue against a possible “separate effect
of NK cell depletion” on the LCMV-specific CD8⁺ T cell response in *Trail*^{-/-} versus WT mice.

4. Lastly, as an additional remark to this comment: In our reply to major comment #2 of
Reviewer #3 further below, we provide data supporting the conclusion that while TRAIL

directly regulates NK cell function during inflammation, its immunomodulatory role does not
 always result in an altered CD8⁺ T cell response. In other terms, upon infection with the
 Armstrong strain of LCMV, the specific CD8⁺ T cell response in *Trail*^{-/-} mice remains
 unaffected, while it is increased upon infection with LCMV WE (**PBP-Figure 7**). Therefore,
 the consistent “TRAIL effect” on NK cells may or may not affect the specific CD8⁺ T cell
 response, in dependence of the nature of the infecting pathogen.

*3. More precision and consistency are needed in Results/Methods/Legends regarding whether*
 *total splenocyte or isolated NK cells are assayed.*

We have improved this by adding more information in the figure legends and the methods of
 the revised manuscript. We also added some modifications to the result part to avoid possible
 confusion on whether whole splenocytes or isolated NK cells were tested.

Importantly, we show now that the signaling downstream of IL-15 receptor is impaired in *Trail*
 649 ^{-/-} versus WT NK cells whether these cells are analyzed after *in vitro* IL-15 stimulation of whole
 splenocytes or whether these cells are first MACS-purified before *in vitro* IL-15 stimulation
 and analysis (Figure 5; Figure 5—figure supplement 1A)

*4. In Fig 1C, titers are inappropriately presented and statistics incorrectly applied for day 12.*
 *An undetectable virus load cannot be accurately displayed as 0, but should instead be assigned*
 *a less-than value just below the L.O.D. Your statistics should be recalculated with these values.*
 *Also, not clear if the "***" in spleen viral load plot is incorrectly labeling day 8 when it should*
 *be day 7.*

The Figure 1C has been corrected accordingly and is displayed in the new version of the
 manuscript.

We addressed the issue on statistics in our reply to minor comment #1 of Reviewer #1 (see
 above).

*5. In Figure 2C, legend should reflect whether these histograms are gated on NK cells (as*
 *described in results). Either the results or legend need to be edited.*

Figure 2C shows TRAIL and TRAIL-R (DR5) surface protein expression on NK cells. This
 information has been added to the figure legend of the revised manuscript.

6. *The Tnfsf10^{-/-} mice were originally made in 129 stem cells, such that the low Ly49H*
*expression could reflect incomplete backcrossing along chromosome 6. The authors should*
*comment on this and any other potential effects of 129 gene carryover that might impact their*
*measurements. This possibility may negate many of the arguments discussed in the 2nd*
*paragraph of the discussion.*

*Tnfsf10/Trail^{-/-} mice were previously used at our Institute (35) and have been further*
*backcrossed to a C57BL/6 genetic background for a total of at least 10 times before generating*
*the data presented in our manuscript. We have indicated this information in the Materials and*
*Methods, in the section on “mice”.*

The targeted *Tnfsf10/Trail* locus in *Trail^{-/-}* mice is located on mouse chromosome 3, while the
*Klra8/Ly49h* locus is positioned on chromosome 6. Therefore, while the observed low Ly49H
expression may reflect incomplete backcrossing, we think it is unlikely that this would have
happened specifically on chromosome 6 after ≥ 10 rounds of backcrossing.

Please, note that Reviewer #3 also inquired about the genetic background of the *Trail^{-/-}* mice
used in our study (Reviewer #3, major comment #1).

7. *In Figure 4, do E:T reflect "splenocyte" to target or "calculated NK cell within splenocyte"*
*to target ratios?*

The data in Figure 4A show "calculated NK cell within splenocyte" to target ratios; this
information has been accordingly specified in the new version of our manuscript, in the
Materials and Methods, in the section on “NK cell cytotoxicity assays”.

**Comments of Reviewer #3:**688 **Reviewer #3 / Major comments:**

*1. Based on the available information (Reference #57), TRAIL-deficient mice used in this work*
*were initially produced on non-B6 background, then backcrossed only five times to B6 mice.*
*The manuscript indicates "All mice were maintained on a C57BL/6 genetic background",*
*which needs to be further clarified. The genetic background of TRAIL-deficient mice and WT*
*mice should be extensively matched to exclude the possible non-TRAIL-related effects.*

This relevant comment has been addressed in our reply to minor comment #6 of Reviewer #2.
In addition, the information on the genetic background of *Trail*^{-/-} mice is now indicated in the
Materials and Methods, in the section on “mice”.

*2. A previous study (Badovinac et al, 2006, not cited) reported that there was no difference in*
*LCMV specific CD8⁺ T cell responses, including GP33-41-specific response, between TRAIL-*
*deficient mice (backcrossed 10 times to B6) and WT B6 mice. This discrepancy should be*
*discussed.*

Badovinac *et al.* used the Armstrong (ARM) strain of LCMV as a model of infection (36),
while we used LCMV-WE for our study. To compare the effect of *Trail* deficiency for the
LCMV-specific CD8⁺ T cell response with these different strains of virus, we transferred
congenic (*Trail*^{+/+}) T cell receptor (TCR) transgenic CD8⁺ T cells specific for the LCMV
glycoprotein GP₃₃₋₄₁ (P14 cells) into WT and *Trail*^{-/-} mice previously infected with a same dose
of LCMV-WE and LCMV-ARM virus (**PBP-Figure 7A**). Under these conditions, a difference
in the expansion of P14 cells in *Trail*^{-/-} versus WT mice was only observed after infection with
LCMV-WE (**PBP-Figure 7B**). Therefore, these data are in agreement with the study of
Badovinac *et al.*

We believe that these *Trail*-dependent differences in the T cell response with LCMV-WE
compared to LCMV-ARM virus are caused by distinct expansion magnitude or kinetics of
virus-specific CD8⁺ T cells, which are associated with different kinetics of virus replication
between LCMV-WE and LCMV-ARM strains (37).

We present and discuss these findings in the revised manuscript.

Of note, there are conflicting data on the contribution of NK cells to the antiviral CD8⁺ T cell
 response during LCMV-ARM infection, with reports mentioning no role for NK cells (24, 38)
 and other studies finding an enhanced specific CD8⁺ T cell response in NK cell-depleted mice
 infected with this LCMV strain (6).

**PBP-Figure 7. *Trail*-dependent expansion of virus-specific CD8⁺ T cells is LCMV strain-specific.**
 **(A)** Experimental setup of P14 cell transfer experiments following infection of WT and *Trail*^{-/-} mice
 with the indicated LCMV strains. **(B)** P14 cell expansion was analyzed in spleen 4 days post infection
 with 10⁵ plaque-forming units (pfu) of LCMV (n=5-6 mice per group of infected mice). Data indicate
 mean ± SEM. Statistical analyses were performed using unpaired two-tailed *t* test. *p<0.05.

To address the general relevance of the non-canonical contribution of *Trail* for the modulation
 of NK cell function *in vivo*, we also challenged *Trail*^{-/-} and WT mice with polyinosinic-
 polycytidylic acid (poly(I:C)), a synthetic analog of double-stranded RNA that mimics viral
 infection. In line with the data generated during LCMV infection, the signaling downstream of
 IL-15 receptor was also diminished in *Trail*^{-/-} versus WT NK cells in this model of sterile
 inflammation (**PBP-Figure 8**).

Therefore, the immunomodulatory function of *Trail* for NK cells is independent of the infection
 with specific LCMV strains. However, the overall effect of this regulatory mechanisms on
 CD8⁺ T cell response appears to be LCMV strain-dependent. Yet, while the effect on the virus-
 specific CD8⁺ T cell response is a consequence of the altered NK cell cytotoxicity in LCMV-
 infected *Trail*^{-/-} mice, our study rather focuses on the non-canonical, regulatory role of TRAIL
 signaling for NK cell function during inflammation.

**PBP-Figure 8. *Trail* regulates NK cells also during sterile inflammation *in vivo*.** Indicated groups
 of mice were injected i.p. with 150µg of polyinosinic-polycytidylic acid (poly(I:C)) and flow cytometry
 was applied on splenic NK cells to assess S6 phosphorylation after 14 hours. Data indicate mean ± SEM
 (n=3 per group of poly(I:C) injected mice). Statistical analyses were performed using unpaired two-
 tailed *t* test. *p<0.05.

*3. Abstract indicates "Our study reveals...immunoregulatory role of TRAIL signaling on NK*
 *cells for the granzyme B-dependent elimination of antiviral T cells", but no clear evidence to*
 *support this claim was presented in the manuscript. Is there any evidence that TRAIL-deficient*
 *NK and WT NK cells kill directly and differentially LCMV-specific T cells? If not, this could*
 *be misleading and should be rephrased. Additionally, possible direct interactions (e.g., via*
 *activating receptors) between NK and T cells should be discussed.*

- Note that a similar comment on the killing of virus-specific CD8⁺ T cells by NK cells was
 raised by Reviewer #1 (major comment #1 and #3) and Reviewer #2 (major comment #4). This
 issue was addressed above, as a reply to the major comment # 4 of Reviewer #2, and in Figure
 4C of the new version of the manuscript, which shows that *Trail*^{-/-} NK cells have an impaired
 ability to eliminate LCMV-specific CD8⁺ T cells *in vivo*, compared to WT NK cells. Our data
 suggest that this phenotype is caused by a reduced ability of activated *Trail*^{-/-} NK cells to
 mediate perforin/granzyme-dependent target cell lysis.

- Previous studies have shown that NK cells recognize activated LCMV-specific T cells via
 interactions between NKG2D (7) or NKp46/NCR1 (8) on NK cells and their respective ligands
 on target T cells. Since we did not find differences in the expression of these and other possibly
 relevant receptors on *Trail*^{-/-} versus WT NK cells during LCMV infection (Figure 4—figure
 supplement 1), we infer in the discussion that "...TRAIL has only minor effects – if any – on
 NK cell development and that it likely does not alter the ability of NK cells to recognize
 activated T cells".

4. Examination of granzyme B expression, pAKT/pS6 and NK1.1-induced IFN γ
production following adoptive transfer (eg, Fig. 7D; and transfer of TRAIL-deficient spleen
cells into WT mice) may yield informative data. NK1.1-crosslinking experiments were
performed on MACS-purified DX5⁺ cells (Fig. 5 Suppl. 3D), but more accurate data might be
produced with whole splenocytes with less complications.

- A similar comment on GZMB and IFN γ expression after adoptive transfer (in the
experimental setup showed in Figure 7C and D) was raised by Reviewer #2, which we
addressed above (Reviewer #2, major comment #3).

- We MACS-purified NK cells (based on positive selection of CD49B/DX5⁺ cells) to minimize
a possible influence on these cells by activated NKT cells. Indeed, NKT cells also express
NK1.1 and produce IFN γ upon crosslinking of NK1.1 (39, 40). Yet it is also true that some
NKT cells also express CD49B (41). Nevertheless, flow cytometry data in Figure 5—figure
supplement 2D show *bona fide* NK cells and exclude CD3⁺ NKT cells.

We have also performed NK1.1 crosslinking experiments on whole splenocytes, yet could not
see activation of NK cells in such conditions (data not shown).

**References**

- 1. Paolini, R., Bernardini, G., Molfetta, R. & Santoni, A. 2015. NK cells and interferons.
*Cytokine Growth Factor Rev*, **26**:113-20. 10.1016/j.cytogfr.2014.11.003
- 2. Ou, R., Zhou, S., Huang, L. & Moskophidis, D. 2001. Critical role for alpha/beta and gamma
interferons in persistence of lymphocytic choriomeningitis virus by clonal exhaustion of
cytotoxic T cells. *J Virol*, **75**:8407-23.
- 3. Bartholdy, C., Christensen, J. P., Wodarz, D. & Thomsen, A. R. 2000. Persistent virus
infection despite chronic cytotoxic T-lymphocyte activation in gamma interferon-deficient
mice infected with lymphocytic choriomeningitis virus. *J Virol*, **74**:10304-11.
- 4. Moskophidis, D., Bategay, M., Bruendler, M. A., Laine, E., Gresser, I. & Zinkernagel, R.
791 M. 1994. Resistance of lymphocytic choriomeningitis virus to alpha/beta interferon and to
792 gamma interferon. *J Virol*, **68**:1951-5.
- 5. Wille, A., Gessner, A., Lothar, H. & Lehmann-Grube, F. 1989. Mechanism of recovery from
acute virus infection. VIII. Treatment of lymphocytic choriomeningitis virus-infected mice
with anti-interferon-gamma monoclonal antibody blocks generation of virus-specific
cytotoxic T lymphocytes and virus elimination. *Eur J Immunol*, **19**:1283-8.
10.1002/eji.1830190720
- 6. Waggoner, S. N., Cornberg, M., Selin, L. K. & Welsh, R. M. 2011. Natural killer cells act
as rheostats modulating antiviral T cells. *Nature*, **481**:394-8. 10.1038/nature10624
- 7. Lang, P. A., Lang, K. S., Xu, H. C., Grusdat, M., Parish, I. A., Recher, M., Elford, A. R.,
Dhanji, S., Shaabani, N., Tran, C. W., Dissanayake, D., Rahbar, R., Ghazarian, M., Brustle,
802 A., Fine, J., Chen, P., Weaver, C. T., Klose, C., Diefenbach, A., Haussinger, D., Carlyle, J.
R., Kaech, S. M., Mak, T. W. & Ohashi, P. S. 2012. Natural killer cell activation enhances
immune pathology and promotes chronic infection by limiting CD8⁺ T-cell immunity. *Proc*
*Natl Acad Sci U S A*, **109**:1210-5. 10.1073/pnas.1118834109
- 8. Crouse, J., Bedenikovic, G., Wiesel, M., Ibberson, M., Xenarios, I., Von Laer, D., Kalinke,
U., Vivier, E., Jonjic, S. & Oxenius, A. 2014. Type I interferons protect T cells against NK
cell attack mediated by the activating receptor NCR1. *Immunity*, **40**:961-73.
10.1016/j.immuni.2014.05.003
- 9. Xu, H. C., Grusdat, M., Pandyra, A. A., Polz, R., Huang, J., Sharma, P., Deenen, R., Kohrer,
811 K., Rahbar, R., Diefenbach, A., Gibbert, K., Lohning, M., Hocker, L., Waibler, Z.,
Haussinger, D., Mak, T. W., Ohashi, P. S., Lang, K. S. & Lang, P. A. 2014. Type I interferon

- protects antiviral CD8⁺ T cells from NK cell cytotoxicity. *Immunity*, **40**:949-60.
 10.1016/j.immuni.2014.05.004
- 10. Azijli, K., Weyhenmeyer, B., Peters, G. J., De Jong, S. & Kruyt, F. A. 2013. Non-
 canonical kinase signaling by the death ligand TRAIL in cancer cells: discord in the death
 receptor family. *Cell Death Differ*, **20**:858-68. 10.1038/cdd.2013.28
- 11. Refaat, A., Abd-Rabou, A. & Reda, A. 2014. TRAIL combinations: The new 'trail' for
 cancer therapy (Review). *Oncol Lett*, **7**:1327-1332. 10.3892/ol.2014.1922
- 12. Zauli, G., Sancilio, S., Cataldi, A., Sabatini, N., Bosco, D. & Di Pietro, R. 2005. PI-
 3K/Akt and NF-kappaB/IkappaBalpha pathways are activated in Jurkat T cells in response
 to TRAIL treatment. *J Cell Physiol*, **202**:900-11. 10.1002/jcp.20202
- 13. Xu, J., Zhou, J. Y., Wei, W. Z. & Wu, G. S. 2010. Activation of the Akt survival
 pathway contributes to TRAIL resistance in cancer cells. *PLoS One*, **5**:e10226.
 10.1371/journal.pone.0010226
- 14. Azijli, K., Yuvaraj, S., Peppelenbosch, M. P., Wurdinger, T., Dekker, H., Joore, J., Van
 Dijk, E., Quax, W. J., Peters, G. J., De Jong, S. & Kruyt, F. A. 2012. Kinome profiling of
 non-canonical TRAIL signaling reveals RIP1-Src-STAT3-dependent invasion in resistant
 non-small cell lung cancer cells. *J Cell Sci*, **125**:4651-61. 10.1242/jcs.109587
- 15. Song, J. J., Kim, J. H., Sun, B. K., Alcalá, M. A., Jr., Bartlett, D. L. & Lee, Y. J. 2010.
 c-Cbl acts as a mediator of Src-induced activation of the PI3K-Akt signal transduction
 pathway during TRAIL treatment. *Cell Signal*, **22**:377-85. 10.1016/j.cellsig.2009.10.007
- 16. Secchiero, P., Gonelli, A., Carnevale, E., Milani, D., Pandolfi, A., Zella, D. & Zauli, G.
 2003. TRAIL promotes the survival and proliferation of primary human vascular endothelial
 cells by activating the Akt and ERK pathways. *Circulation*, **107**:2250-6.
 10.1161/01.CIR.0000062702.60708.C4
- 17. Falschlehner, C., Emmerich, C. H., Gerlach, B. & Walczak, H. 2007. TRAIL signalling:
 decisions between life and death. *Int J Biochem Cell Biol*, **39**:1462-75.
 10.1016/j.biocel.2007.02.007
- 18. Marcais, A., Viel, S., Grau, M., Henry, T., Marvel, J. & Walzer, T. 2013. Regulation
 of mouse NK cell development and function by cytokines. *Front Immunol*, **4**:450.
 10.3389/fimmu.2013.00450
- 19. Marcais, A. & Walzer, T. 2014. mTOR: a gate to NK cell maturation and activation.
 *Cell Cycle*, **13**:3315-6. 10.4161/15384101.2014.972919
- 20. Marcais, A., Cherfils-Vicini, J., Viant, C., Degouve, S., Viel, S., Fenis, A., Rabilloud,
 846 J., Mayol, K., Tavares, A., Bienvenu, J., Gangloff, Y. G., Gilson, E., Vivier, E. & Walzer,

- 847 T. 2014. The metabolic checkpoint kinase mTOR is essential for IL-15 signaling during the
848 development and activation of NK cells. *Nat Immunol*, **15**:749-757. 10.1038/ni.2936
- 21. Nandagopal, N., Ali, A. K., Komal, A. K. & Lee, S. H. 2014. The Critical Role of IL-
15-PI3K-mTOR Pathway in Natural Killer Cell Effector Functions. *Front Immunol*, **5**:187.
10.3389/fimmu.2014.00187
- 22. Ali, A. K., Nandagopal, N. & Lee, S. H. 2015. IL-15-PI3K-AKT-mTOR: A Critical
Pathway in the Life Journey of Natural Killer Cells. *Front Immunol*, **6**:355.
10.3389/fimmu.2015.00355
- 23. Kayagaki, N., Yamaguchi, N., Nakayama, M., Takeda, K., Akiba, H., Tsutsui, H.,
Okamura, H., Nakanishi, K., Okumura, K. & Yagita, H. 1999. Expression and function of
TNF-related apoptosis-inducing ligand on murine activated NK cells. *J Immunol*, **163**:1906-
13.
- 24. Cook, K. D. & Whitmire, J. K. 2013. The depletion of NK cells prevents T cell
exhaustion to efficiently control disseminating virus infection. *J Immunol*, **190**:641-9.
10.4049/jimmunol.1202448
- 25. Shigeno, M., Nakao, K., Ichikawa, T., Suzuki, K., Kawakami, A., Abiru, S., Miyazoe,
S., Nakagawa, Y., Ishikawa, H., Hamasaki, K., Nakata, K., Ishii, N. & Eguchi, K. 2003.
Interferon-alpha sensitizes human hepatoma cells to TRAIL-induced apoptosis through
DR5 upregulation and NF-kappa B inactivation. *Oncogene*, **22**:1653-62.
10.1038/sj.onc.1206139
- 26. Voskoboinik, I., Whisstock, J. C. & Trapani, J. A. 2015. Perforin and granzymes:
function, dysfunction and human pathology. *Nat Rev Immunol*, **15**:388-400.
10.1038/nri3839
- 27. Mack, E. A., Kallal, L. E., Demers, D. A. & Biron, C. A. 2011. Type 1 interferon
induction of natural killer cell gamma interferon production for defense during lymphocytic
choriomeningitis virus infection. *MBio*, **2**: 10.1128/mBio.00169-11
- 28. Utermohlen, O., Dangel, A., Tarnok, A. & Lehmann-Grube, F. 1996. Modulation by
gamma interferon of antiviral cell-mediated immune responses in vivo. *J Virol*, **70**:1521-6.
- 29. Pien, G. C., Nguyen, K. B., Malmgaard, L., Satoskar, A. R. & Biron, C. A. 2002. A
unique mechanism for innate cytokine promotion of T cell responses to viral infections. *J*
*Immunol*, **169**:5827-37.
- 30. Tewari, K., Nakayama, Y. & Suresh, M. 2007. Role of direct effects of IFN-gamma on
T cells in the regulation of CD8 T cell homeostasis. *J Immunol*, **179**:2115-25.

- 31. Koo, G. C. & Peppard, J. R. 1984. Establishment of monoclonal anti-Nk-1.1 antibody.
*Hybridoma*, **3**:301-3. 10.1089/hyb.1984.3.301
- 32. Wang, X., Li, H., Matte-Martone, C., Cui, W., Li, N., Tan, H. S., Roopenian, D. &
Shlomchik, W. D. 2011. Mechanisms of antigen presentation to T cells in murine graft-
versus-host disease: cross-presentation and the appearance of cross-presentation. *Blood*,
**118**:6426-37. 10.1182/blood-2011-06-358747
- 33. Krebs, P., Barnes, M. J., Lampe, K., Whitley, K., Bahjat, K. S., Beutler, B., Janssen, E.
& Hoebe, K. 2009. NK-cell-mediated killing of target cells triggers robust antigen-specific
T-cell-mediated and humoral responses. *Blood*, **113**:6593-602. 10.1182/blood-2009-01-
201467
- 34. Coleman, M. A., Bridge, J. A., Lane, S. W., Dixon, C. M., Hill, G. R., Wells, J. W.,
Thomas, R. & Steptoe, R. J. 2013. Tolerance induction with gene-modified stem cells and
immune-preserving conditioning in primed mice: restricting antigen to differentiated
antigen-presenting cells permits efficacy. *Blood*, **121**:1049-58. 10.1182/blood-2012-06-
434100
- 35. Corazza, N., Jakob, S., Schaer, C., Frese, S., Keogh, A., Stroka, D., Kassahn, D.,
Torgler, R., Mueller, C., Schneider, P. & Brunner, T. 2006. TRAIL receptor-mediated JNK
activation and Bim phosphorylation critically regulate Fas-mediated liver damage and
lethality. *J Clin Invest*, **116**:2493-9. 10.1172/JCI27726
- 36. Badovinac, V. P., Messingham, K. A., Griffith, T. S. & Harty, J. T. 2006. TRAIL
deficiency delays, but does not prevent, erosion in the quality of "helpless" memory CD8 T
cells. *J Immunol*, **177**:999-1006.
- 37. Bocharov, G., Ludewig, B., Bertoletti, A., Klenerman, P., Junt, T., Krebs, P.,
Luzyanina, T., Fraser, C. & Anderson, R. M. 2004. Underwhelming the immune response:
effect of slow virus growth on CD8⁺-T-lymphocyte responses. *J Virol*, **78**:2247-54.
- 38. Su, H. C., Nguyen, K. B., Salazar-Mather, T. P., Ruzek, M. C., Dalod, M. Y. & Biron,
C. A. 2001. NK cell functions restrain T cell responses during viral infections. *Eur J*
*Immunol*, **31**:3048-55. 10.1002/1521-4141(2001010)31:10<3048::AID-
IMMU3048>3.0.CO;2-1
- 39. Arase, H., Arase, N. & Saito, T. 1996. Interferon gamma production by natural killer
(NK) cells and NK1.1⁺ T cells upon NKR-P1 cross-linking. *J Exp Med*, **183**:2391-6.
- 40. Hansen, D. S., Siomos, M. A., Buckingham, L., Scalzo, A. A. & Schofield, L. 2003.
Regulation of murine cerebral malaria pathogenesis by CD1d-restricted NKT cells and the
natural killer complex. *Immunity*, **18**:391-402.

- 41. Pellicci, D. G., Hammond, K. J., Coquet, J., Kyparissoudis, K., Brooks, A. G.,
Kedzierska, K., Keating, R., Turner, S., Berzins, S., Smyth, M. J. & Godfrey, D. I. 2005.
DX5/CD49b-positive T cells are not synonymous with CD1d-dependent NKT cells. *J*
*Immunol*, **175**:4416-25.

Thank you for the transfer of your revised research manuscript to EMBO reports. We have now received the reports from the three referees that were asked to evaluate your revised manuscript, which can be found at the end of this email.

As you will see, all three referees support the publication of your manuscript in EMBO reports, and indicate the concerns raised during a previous round of peer review at a journal outside EMBO press have been adequately addressed.

Before we can proceed with formal acceptance, several editorial points need to be addressed in a final revised version of the manuscript. Please also carefully review the instructions that follow below. Failure to include requested items will delay the acceptance and publication of your study.

When submitting your final revised manuscript, we will require:

1) a .docx formatted version of the final manuscript text (including legends for main figures, EV figures and tables), but without the figures included. Please make sure that the changes are highlighted to be clearly visible. Figure legends should be compiled at the end of the manuscript text.

2) individual production quality figure files as .eps, .tif, .jpg (one file per figure), of main figures and EV figures. Please upload these as separate, individual files upon re-submission.

The Expanded View format, which will be displayed in the main HTML of the paper in a collapsible format, has replaced the Supplementary information. You can submit up to 5 images as Expanded View. Please follow the nomenclature Figure EV1, Figure EV2 etc. The figure legend for these should be included in the main manuscript document file in a section called Expanded View Figure Legends after the main Figure Legends section. Additional Supplementary material should be supplied as a single pdf labeled Appendix. The Appendix should have page numbers and needs to include a table of content on the first page (with page numbers) and legends for all content. Please follow the nomenclature Appendix Figure Sx, Appendix Table Sx etc. throughout the text, and also label the figures and tables according to this nomenclature, and update all call-outs accordingly.

For more details please refer to our guide to authors:

See also our guide for figure preparation:

3) a complete author checklist, which you can download from our author guidelines (<https://www.embopress.org/page/journal/14693178/authorguide>). Please insert page numbers in the checklist to indicate where the requested information can be found in the manuscript. The completed author checklist will also be part of the RPF.

Please also follow our guidelines for the use of living organisms, and the respective reporting guidelines: <http://www.embopress.org/page/journal/14693178/authorguide#livingorganisms>

4) that primary datasets produced in this study (e.g. the RNA-seq. data) are deposited in an appropriate public database. See: <http://embor.embopress.org/authorguide#datadeposition>

The accession numbers and database should be listed in a formal "Data Availability " section (placed after Materials & Methods) that follows the model below. Please note that the Data Availability Section is restricted to new primary data that are part of this study.

Data availability

5) We strongly encourage the publication of original source data with the aim of making primary data more accessible and transparent to the reader. The source data will be published in a separate source data file online along with the accepted manuscript and will be linked to the relevant figure. If you would like to use this opportunity, please submit the source data (for example scans of entire gels or blots, data points of graphs in an excel sheet, additional images, etc.) of your key experiments together with the revised manuscript. If you want to provide source data, please include size markers for scans of entire gels, label the scans with figure and panel number, and send one PDF file per figure.

6) Our journal encourages inclusion of **data citations in the reference list** to directly cite datasets that were re-used and obtained from public databases. Data citations in the article text are distinct from normal bibliographical citations and should directly link to the database records from which the data can be accessed. In the main text, data citations are formatted as follows: "Data ref: Smith et al, 2001" or "Data ref: NCBI Sequence Read Archive PRJNA342805, 2017". In the Reference list, data citations must be labeled with "[DATASET]". A data reference must provide the database name, accession number/identifiers and a resolvable link to the landing page from which the data can be accessed at the end of the reference. Further instructions are available at: <http://www.embopress.org/page/journal/14693178/authorguide#referencesformat>

7) Regarding data quantification and statistics, can you please specify, where applicable, the number "n" for how many independent experiments (biological replicates) were performed, the bars and error bars (e.g. SEM, SD) and the test used to calculate p-values in the respective figure legends. Please provide statistical testing where applicable, and also add a paragraph detailing this to the methods section. See: <http://www.embopress.org/page/journal/14693178/authorguide#statisticalanalysis>

8) Per journal policy, we do not allow 'data not shown'. All data referred to in the paper should be displayed in the main or Expanded View figures, or the Appendix. Thus, please add these data (or change the text accordingly, if these data are not important). See: <http://embor.embopress.org/authorguide#unpublisheddata>

9) Please add up to five key words to the title page.

10) Please add a paragraph named 'Author contributions' to the manuscript text (before the acknowledgements) that defines the contributions of each author.

11) Please add the funding information also directly into our submission system upon re-submission.

In addition I would need from you:

- a short, two-sentence summary of the manuscript
- two to three bullet points highlighting the key findings of your study
- a schematic summary figure (in jpeg or tiff format with the exact width of 550 pixels and a height of not more than 400 pixels) that can be used as a visual synopsis on our website.

I look forward to seeing a revised version of your manuscript when it is ready. Please let me know if you have questions or comments regarding the revision.

REFEREE REPORTS

Referee #1:

The authors present a revised version of their intriguing description of an NK-cell extrinsic role of TRAIL in engaging TRAIL-R on NK cells to support IL-15 triggering of GzmB expression. The consequences of this are increased IFN-g and suppression of the CD8 T cell response against the virus. This is a new mechanism of broad interest to those aiming to understand immunoregulatory functions of NK cells and the environmental mechanisms contributing to functionality of NK cells during inflammation. The authors' additional experiments to address reviewer criticisms and improve the readability of the paper were outstanding. Sufficient information is present in a logical fashion for any reader to appreciate and critique the validity of the results. Some minor curiosities remain, but any more data would overencumber the manuscript and therefore these points should be left for subsequent manuscripts. I congratulate the authors and look forward to seeing this manuscript in print.

Referee #2:

All of my comments have been adequately addressed.

Referee #3:

This manuscript has already been extensively reviewed. I believe that the authors have satisfactorily addressed most of the concerns of the previous reviewers. For technical reasons some of the experiments attempted to address the reviewers' criticisms were not possible or not entirely conclusive. However, on the whole the authors provide sufficient, good quality evidence in my opinion to support their conclusions that TRAIL has a non-apoptotic function via non canonical signaling to enhance IL-15 signaling, thereby promoting enhanced Granzyme B content and cytotoxicity while simultaneously down regulating IFN gamma. I found the revised manuscript to be clear and easy to understand and feel that it merits publication in its present form.

1st Revision - authors' response

24 September 2019

The authors addressed the editorial requests.

2nd Editorial Decision

8 October 2019

Thank you for the submission of your revised manuscript to our editorial offices. Before we can proceed with formal acceptance, I have these few editorial requests that need to be addressed:

- Please remove the synopsis blurb and the bullet points from the main manuscript text. I have saved these separately, and will forward them to our publisher after acceptance of the study.
- Panel E is mentioned twice in the legend of Fig. 2. Please check.
- Please call out the single panels in Fig. EV2 in the manuscript text.
- Please upload the source data for Figs. EV2 and EV3 in separate files (named Source Data Figure EV2 and Source Data Figure EV3). These files will be linked to the respective figures online, and thus need to be separate. Please also call these files out separately, if mentioned in the manuscript text.
- The writing in the synopsis image if sized as it will appear online (see attached) is partially rather

small. Please provide the synopsis image with bigger fonts.

- Finally, please find attached a word file of the manuscript text (provided by our publisher) with changes we ask you to include in your final manuscript text, and some queries, we ask you to address. Please provide your final manuscript file with track changes, in order that we can see the modifications done.

2nd Revision - authors' response

16 October 2019

The authors addressed the final editorial requests.

Corresponding Author Name: Philippe Krebs

Journal Submitted to: EMBO reports

Manuscript Number: EMBOR-2019-48789-T